# Exact Response Theory for Time-Dependent and Stochastic Perturbations

**DOI:** 10.3390/e26010012

**Published:** 2023-12-21

**Authors:** Leonardo Iannella, Lamberto Rondoni

**Affiliations:** 1Dipartimento di Fisica, Università di Torino, Via Pietro Giuria 1, 10125 Torino, Italy; leonardo.iannella69@edu.unito.it; 2Dipartimento di Scienze Matematiche, Politecnico di Torino, Corso Duca Degli Abruzzi 24, 10129 Torino, Italy; 3INFN, Sezione di Torino, Via Pietro Giuria 1, 10125 Torino, Italy

**Keywords:** dynamical systems, probability distributions, observables

## Abstract

The exact, non perturbative, response theory developed within the field of non-equilibrium molecular dynamics, also known as TTCF (transient time correlation function), applies to quite general dynamical systems. Its key element is called the *dissipation function* because it represents the power dissipated by external fields acting on the particle system of interest, whose coupling with the environment is given by deterministic thermostats. This theory has been initially developed for time-independent external perturbations, and then it has been extended to time-dependent perturbations. It has also been applied to dynamical systems of different nature, and to oscillator models undergoing phase transitions, which cannot be treated with, e.g., linear response theory. The present work includes time-dependent stochastic perturbations in the theory using the Karhunen–Loève theorem. This leads to three different investigations of a given process. In the first, a single realization of the stochastic coefficients is fixed, and averages are taken only over the initial conditions, as in a deterministic process. In the second, the initial condition is fixed, and averages are taken with respect to the distribution of stochastic coefficients. In the last investigation, one averages over both initial conditions and stochastic coefficients. We conclude by illustrating the applicability of the resulting exact response theory with simple examples.

## 1. Introduction

Linear response theory successfully describes the behavior of macroscopic systems that are close to thermodynamic equilibrium, obtaining the corresponding transport coefficients by solely using the equilibrium correlation functions of the microscopic fluctuating currents, computed with the equilibrium dynamics, cf. Refs. [1,2,3]. While the range of applicability of the linear theory covers most phenomena occurring on the scale of our daily life, extending well beyond that scale, contemporary science and technology concern scales that are much smaller or that involve extremely large driving forces, which often exceed the linear regime. Indeed, anomalous phenomena, for which the linear transport coefficients do not exist or vanish, are common at the nanometric scale. Strong drivings, such as high voltages, produce large currents that alter the physical properties of the system at hand, leading to non-linear effects. Under external drivings, microscopic motions may be impaired to the point of producing localization or, in any case, drastic changes of states, like phase transitions. In many of these situations, dynamical correlations persist in time and give rise to behaviors that are not fully understood.

Within the field of non-equilibrium molecular dynamics [4], particularly following the discovery of the fluctuation relation, response theory has been generalized so that small systems as well as large drivings can be treated. A general exact (non-perturbative) response theory, also known as TTCF (transient time correlation function), has been derived [4], which has proven very effective. In particular, recently it has allowed treating hard non-equilibrium problems, at low drivings [5,6], drastically improving the signal-to-noise ratio, and providing a superior method with respect to direct averaging for such calculations. Moreover, this theory has allowed the derivation of a host of relations concerning non-equilibrium systems; cf. e.g., Ref. [7]. The key ingredient of this exact response theory is known as the *dissipation function*, first explicitly defined by Evans and Searles as the energy dissipation rate that verifies the fluctuation relation [8,9]. Through this, the quantities singularly used (until then) in specific instances of non-equilibrium molecular dynamic studies have been unified into a single general concept, which can be used in the analysis of any dynamical system. The original result, denoted as the dissipation theorem in Ref. [10], concerned systems subjected to time-independent perturbations, such as an external constant field. Periodic perturbations were then considered by Petravic and Evans, in Refs. [11,12], for time-periodic planar shear flow, and by Todd, in Ref. [13], for time-periodic planar elongational flow.

Inspired by the original works, and profiting from the general applicability of the dissipation function previously evidenced within the framework of the fluctuation relation [14], an extension to the general dynamical systems perturbed by time-dependent vector fields, expressed by the Fourier series, was developed in Ref. [15]. This was conducted on an enlarged phase space, in which the equations of motion are autonomous. Apart from the generality that allows the use of the TTCF outside the realm of molecular dynamics, anywhere dynamical systems are used, our approach preserves the time reversibility of the unperturbed dynamics, which is a fundamental ingredient in many statistical mechanics calculations, including the Onsager reciprocal relations and the fluctuation relations [2,16,17]. Moreover, expressing a signal in terms of the Fourier mode makes the contributions to the responses of various harmonics of the forcing directly computable, and provides important information on both the forcing and response in terms of power spectra, as often required in statistical physics [4,18,19,20,21,22,23,24,25,26,27]. Now, physics applications often deal with periodic functions whose periods exceed any relevant time scale, as in standard approaches to the fluctuation–dissipation relation [1,2,28]. However, the Fourier series formalism naturally lends itself to the analysis of non-periodic time-dependent driving, letting the period of periodic signals grow without bounds. In that case, the Fourier series turns, under relatively mild conditions, into a Fourier transform, which is mathematically suited to treat non-periodic signals [29,30,31].

In the present paper, we take that approach one step forward, allowing stochastic time-dependent perturbations [32,33], which can be represented as Karhunen–Loève expansions [34,35]. In Section 2, we recall the Karhunen–Loève theorem, from which we obtain the particular representation of the stochastic perturbation of our dynamical systems. Section 3 briefly illustrates the linear response theory, highlighting the role of the initial perturbation, and then we also illustrate the time-independent exact response formalism. Section 4 starts from the equations of motion of the deterministic system of interest, to which we add the time-dependent stochastic perturbation discussed in Section 2. The first step is to make our system autonomous by enlarging the phase space, to accommodate a (Hamiltonian) harmonic oscillator, which embeds in itself the time dependence of the vector field. Then, we redefine the quantities of interest in the enlarged phase space and we distinguish three cases: **(a)** the case in which a single realization of the stochastic process is fixed [36], which yields a deterministic process in which only the initial conditions are random; **(b)** the stochastic case in which the initial condition is fixed and one should only average over the realizations of the stochastic coefficients of the Karhunen–Loève expansion; **(c)** the stochastic case in which averages are taken over both the initial conditions in the phase space and the stochastic coefficients of the process. Finally, we compute the generalized exact response formula for any observable of our system. Section 5 illustrates the theory using simple examples, such as the perturbed harmonic oscillator, and we compare the linear and the exact response results. Section 6 presents this explicit calculation for forcing with stochastic amplitude and a single Fourier component. It reports expressions for a non-equilibrium simple system: one particle in a viscous medium, perturbed by stochastic forcing with infinitely many frequencies and a non-vanishing mean. The various terms of the response theory are analytically expressed as functions of the stochastic coefficients. Generally, quantitative results now necessitate a numerical treatment. In Section 7, we summarize our work, discuss our results, and note how even the standard linear response theory may benefit from the extended phase space approach. Indeed, that allows one to properly treat the initial condition of the time-dependent perturbation, if one only considers finite evolution times, as required in many modern investigations of fluctuating observables. Appendix A computes some integrals used in the main text.

## 2. The Karhunen–Loève Theorem

The Karhunen–Loève theorem allows a stochastic process to be represented as an infinite linear combination of orthogonal functions, analogous to a Fourier series, with stochastic rather than deterministic coefficients. If ${\left\{X_{t}\right\}}_{t\in [a,b]}$ is a centered stochastic process satisfying certain continuity conditions, one can decompose it as a sum of pairwise orthogonal functions multiplied by random coefficients that are pairwise uncorrelated and, hence, orthogonal in probability space. More precisely, the following holds [34]:
**Theorem** **1**(Karhunen–Loève)**.**
*Let $X_{t}$, t∈[a,b] be a square-integrable stochastic process with a zero mean, defined on a probability space (Θ,F,P), with a continuous covariance function $K_{X}\left(s,t\right)$. By letting $e_{k}$ be an orthonormal basis on $L^{2}\left(\left[a,b\right]\right)$ formed by the eigenfunctions of $T_{K_{X}}$ with respective eigenvalues $\lambda_{k}$, $X_{t}$ admits the following representation:*
(1)$$X_{t}=\sum_{k=1}^{\infty }Z_{k}e_{k}\left(t\right),$$
*where the convergence is in $L^{2}$, uniform in t and*
(2)$$Z_{k}=\int_{a}^{b}X_{t}e_{k}\left(t\right)dt.$$
*Furthermore, the random variables $Z_{k}$ have a zero mean, are uncorrelated, and have variance $\lambda_{k}$:*
(3)$$\langle Z_{k}\rangle =0,\;\forall k\in N\;and\;\langle Z_{i}Z_{j}\rangle =\delta_{ij}\lambda_{j},\;\forall i,j\in N.$$
As an example, consider the particular case of a Wiener process, $w_{t}$ with t∈[0,T], and covariance function $K_{w}\left(t,s\right)=\min\left(s,t\right)$. To find the corresponding eigenvalues and eigenvectors, we need to solve the following integral equation:(4)$$\int_{a}^{b}K_{w}\left(s,t\right)e\left(s\right)ds=\int_{0}^{T}\min\left(s,t\right)e\left(s\right)ds=\int_{0}^{t}se\left(s\right)ds+t\int_{t}^{T}e\left(s\right)ds=\lambda e\left(t\right)\;,\;0\leq t\leq T$$
Differentiating once with respect to *t* yields:(5)$$\int_{t}^{T}e\left(s\right)ds=\lambda e^{\prime }\left(t\right)$$
A second differentiation produces:(6)$$-e\left(t\right)=\lambda e^{\prime \prime }\left(t\right).$$
whose general solution takes the form:(7)$$e\left(t\right)=A\sin\left(\frac{t}{\sqrt{\lambda }}\right)+B\cos\left(\frac{t}{\sqrt{\lambda }}\right),$$
where *A* and *B* are determined by the boundary conditions. Setting t=0 in the integral equation gives e(0)=0, which implies B=0; setting t=T in Equation (Equation 5) yields $e^{\prime }\left(T\right)=0$, where:(8)$$\lambda_{k}={\left(\frac{T}{(k-\frac{1}{2})\pi }\right)}^{2},\;k\geq 1.$$
The corresponding eigenfunctions are, thus, given by the following:(9)$$e_{k}\left(t\right)=A\sin\left[\left(k-\frac{1}{2}\right)\pi \frac{t}{T}\right],\;k\geq 1$$
and *A* is finally determined by the normalization condition:(10)$$\int_{0}^{T}e_{k}^{2}\left(t\right)dt=1\;\mathrm{which}\;\mathrm{leads}\;\mathrm{to}\;\;\;A=\sqrt{\frac{2}{T}}$$
Finally, we obtain the following representation of the Wiener process:(11)$$w_{t}=\frac{\sqrt{2T}}{\pi }\sum_{k=1}^{\infty }Z_{k}\frac{\sin\left[\left(k-\frac{1}{2}\right)\pi \frac{t}{T}\right]}{\left(k-\frac{1}{2}\right)}$$
where ${\left\{Z_{k}\right\}}_{k=1}^{\infty }$ is a sequence of independent Gaussian random variables, each having a zero mean and unit variance. This representation is valid for t∈[0,T] for any T>0 and, as stated in the theorem, convergence is in the $L^{2}$ norm and uniform in *t*.

## 3. Response Theory

Linear response theory is suitable to describe the evolution of observables of systems subjected to small perturbations, while the exact response applies regardless of the magnitude of the perturbation. To set our notation, let us recall the main aspects of the time-dependent linear theory [1]. Given a Hamiltonian dynamical system, with Hamiltonian $H_{0}$, consider a Hamiltonian perturbation $H_{p}\left(\Gamma ,t\right)=-F\left(t\right)A\left(\Gamma \right)$, whose small intensity is F, and *A* has the dimensions of energy. The new perturbed Hamiltonian is given by the following:(12)$$H\left(\Gamma ,t\right)=H_{0}\left(\Gamma \right)+H_{p}\left(\Gamma ,t\right)=H_{0}\left(\Gamma \right)-F\left(t\right)A\left(\Gamma \right)$$
and the equations of motion take the following form:(13)$$\begin{array}{ccc} & & \frac{dq_{j}}{dt}=\frac{\partial H_{0}}{\partial p_{j}}-F\left(t\right)\frac{\partial A}{\partial p_{j}}\equiv \frac{\partial H_{0}}{\partial p_{j}}-F\left(t\right)k_{j}^{q}\;,\;\mathrm{with}\;k_{j}^{q}=\frac{\partial A}{\partial p_{j}} \end{array}$$(14)$$\begin{array}{ccc} & & \frac{dp_{j}}{dt}=-\frac{\partial H_{0}}{\partial q_{j}}+F\left(t\right)\frac{\partial A}{\partial q_{j}}\equiv -\frac{\partial H_{0}}{\partial q_{j}}-F\left(t\right)k_{j}^{p}\;,\;\mathrm{with}\;k_{j}^{p}=-\frac{\partial A}{\partial q_{j}} \end{array}$$
By denoting *G* as the corresponding vector field, we have the following:(15)$$G\left(\Gamma ,t\right)=\left(\begin{array}{cc} \; & \frac{\partial H_{0}}{\partial p_{1}}\\ \; & \vdots \\ \; & -\frac{\partial H_{0}}{\partial q_{N}} \end{array}\right)-F\left(t\right)\left(\begin{array}{cc} \; & k_{1}^{q}\\ \; & \vdots \\ \; & k_{N}^{p} \end{array}\right)=G_{0}\left(\Gamma \right)+G_{ext}\left(\Gamma ,t\right)$$
and the Liouville operator
(16)$$\begin{array}{cc} -iL & =-\left(G\cdot \nabla_{\Gamma }+{\mathrm{div}}_{\Gamma }G\right)=-\left(G_{0}\cdot \nabla_{\Gamma }+{\mathrm{div}}_{\Gamma }G_{0}\right)-\left(G_{ext}\cdot \nabla_{\Gamma }+{\mathrm{div}}_{\Gamma }G_{ext}\right)\\ \; & =-i\left(L_{0}+L_{ext}\right). \end{array}$$
allows us to write the continuity equation for the probability distribution in the phase space as follows:(17)$$\partial_{t}f=-iLf=-i\left(L_{0}+L_{ext}\left(t\right)\right)f$$
where
(18)$$iL_{0}f=\left\{f,H_{0}\right\}=\sum_{j=1}^{N}\left(\frac{\partial H_{0}}{\partial p_{j}}\frac{\partial }{\partial q_{j}}-\frac{\partial H_{0}}{\partial q_{j}}\frac{\partial }{\partial p_{j}}\right)f$$
and
(19)$$iL_{ext}f=-F\left(t\right)\left\{f,A\right\}=-F\left(t\right)\sum_{j=1}^{N}\left(k_{j}^{q}\frac{\partial }{\partial q_{j}}+k_{j}^{p}\frac{\partial }{\partial p_{j}}\right)f$$
whose solution can be expressed as follows:(20)$$f_{t}\left(\Gamma \right)=e^{-itL_{0}}f_{0}\left(\Gamma \right)-i\int_{0}^{t}dt^{\prime }e^{-i\left(t-t^{\prime }\right)L_{0}}L_{ext}\left(t^{\prime }\right)f_{t^{\prime }}\left(\Gamma \right)$$
As well-known, this is an exact, but not useful, expression. However, the fact that $L_{ext}$ is proportional to the small intensity of the perturbation F, justifies the following linear approximation of $f_{t}$:(21)$$f_{t}\left(\Gamma \right)=f_{eq}\left(\Gamma \right)-i\int_{0}^{t}dt^{\prime }e^{-i\left(t-t^{\prime }\right)L_{0}}L_{ext}\left(t^{\prime }\right)f_{eq}\left(\Gamma \right)+H.O.$$
where H.O. stands for negligible higher-order terms in F. More than the probabilities, though, we are interested in the evolution of observables, which are identified with their evolving ensemble averages. We denote one observable by O:M→R, and its ensemble average at time *t* by the following:(22)$${\langle O\rangle }_{t}\equiv \int_{M}O\left(\Gamma \right)f_{t}\left(\Gamma \right)d\Gamma$$
In the linear regime, we can then write:(23)$${\langle O\rangle }_{t}-{\langle O\rangle }_{0}=\int_{0}^{t}dt^{\prime }R\left(t-t^{\prime }\right)F\left(t^{\prime }\right)$$
where
(24)$$R\left(t\right)=\beta \int_{M}d\Gamma f_{0}\left(\Gamma \right)\dot{A}\left(\Gamma \right)e^{itL_{0}}O\left(\Gamma \right)=\beta {\langle \dot{A}\left(O\circ S_{0}^{t}\right)\rangle }_{0}$$
is called the response function and subscript 0 in $S_{0}^{t}$ indicates the unperturbed evolution. This is a very important result: The response to a small perturbation is determined by the equilibrium dynamics and by the time correlation function of the perturbation and the observable of interest, computed with respect to the known equilibrium ensemble. For thermodynamic systems, this description is fully satisfactory, because the observables of interest in such systems only negligibly fluctuate, and the observed signal concerning each single object practically equals the ensemble-averaged signal. This is not guaranteed more when non-thermodynamic systems—or large time-dependent perturbations—are considered. Therefore, an extension of the linear response theory to small systems or large perturbations is required to address these two issues.

The (time-independent) exact response theory initially developed in Ref. [10] may be adapted to describe a generic dynamical system, $S^{t}:M\to M$, on a phase space M, where the time *t* may be continuous or discrete [37]. Let us focus on the case in which $S^{t}\Gamma$, with Γ∈M represents the solution at time *t* of the differential equation:(25)$$\dot{\Gamma }=G\left(\Gamma \right)\;\;\mathrm{with}\;\mathrm{initial}\;\mathrm{condition}\;\;\Gamma \in M$$
For any such system, the phase space variation rate, Λ:M→R, is the divergence in M of the vector field:(26)$$\Lambda =\nabla_{\Gamma }\cdot G$$
We denote its time integral along the phase space trajectory segment delimited by the points reached at time *s* and time *t* by the following:(27)$$\Lambda_{s,t}\left(\Gamma \right)=\int_{s}^{t}\Lambda \left(S^{r}\Gamma \right)dr$$
which gives the variation factor of the phase space volume element from $S^{s}\Gamma$ to $S^{t}\Gamma$.

Assume M is endowed with a probability measure $d\mu \left(\Gamma \right)=f_{0}\left(\Gamma \right)d\Gamma$, of density $f_{0}$, which evolves as prescribed by the Liouville equation:(28)$$\frac{\partial f}{\partial t}\left(\Gamma \right)=-G\left(\Gamma \right)\cdot \nabla_{\Gamma }f\left(\Gamma \right)-f\left(\Gamma \right)\nabla_{\Gamma }\cdot G\left(\Gamma \right)$$
We denote by $f_{t}$ the probability density at time *t*, obtained as the solution of Equation (Equation 28) with the initial condition, $f_{0}$. This can be rewritten in terms of the *dissipation function* [38]:(29)$$\Omega^{f_{t}}\left(\Gamma \right)=-G\left(\Gamma \right)\cdot \nabla_{\Gamma }\ln f_{t}(\Gamma )-\Lambda \left(\Gamma \right)$$
as
(30)$$\frac{\partial f_{t}}{\partial t}\left(\Gamma \right)=\Omega^{f_{t}}\left(\Gamma \right)f_{t}\left(\Gamma \right)$$
Starting at a point Γ∈M, the corresponding integral of $\Omega^{f_{0}}$ along the trajectory segment delimited by $S^{s}\Gamma$ and $S^{t}\Gamma$ is given by the following:(31)$$\Omega_{s,t}^{f_{0}}\left(\Gamma \right)=\int_{s}^{t}\Omega^{f_{0}}\left(S^{s}\Gamma \right)ds=\ln\frac{f_{0}\left(S^{s}\Gamma \right)}{f_{0}\left(S^{t}\Gamma \right)}-\int_{s}^{t}\Lambda \left(S^{s}\Gamma \right)ds,$$
and the solution of the Liouville equation can be equivalently expressed as follows:(32)$$f_{s+t}\left(S^{t}\Gamma \right)=\exp\left\{-\Lambda_{0,t}\left(\Gamma \right)\right\}f_{s}\left(\Gamma \right),$$
and as follows:(33)$$f_{s+t}\left(\Gamma \right)=\exp\left\{\Omega_{-t,0}^{f_{s}}\left(\Gamma \right)\right\}f_{s}\left(\Gamma \right)$$
Note that the dynamics are assumed to be invertible, which does not mean that it has to be time-reversal invariant; it suffices that each point in a trajectory has a unique pre-image under the dynamics. The ensemble average of an observable O at time *t*,
(34)$${\langle O\rangle }_{t}=\int O\left(\Gamma \right)f_{t}\left(\Gamma \right)d\Gamma$$
can eventually be expressed as follows:(35)$${\langle O\rangle }_{t}={\langle O\rangle }_{0}+\int_{0}^{t}{\langle \Omega^{f_{0}}\;\left(O\circ S^{s}\right)\rangle }_{0}ds$$
Interestingly, Equation (Equation 35) only requires the unperturbed initial distribution $f_{0}$, a striking analogy to the linear response Formulaes (Equation 23) and (Equation 24), although it is an exact—not an approximate/perturbative—response formula. The difference lies in the fact that, unlike the linear response formulae, here, dynamic $S^{s}$ is the perturbed one. See, e.g., Refs. [4,10,38] for detailed derivations.

For properly chosen $f_{0}$, $\Omega^{f_{0}}$ represents the dissipative flux, like $\dot{A}=\left\{A,H\right\}=\left\{A,H_{0}\right\}$ which appears in Equation (Equation 24). In particular, non-equilibrium molecular dynamics models require $f_{0}$ to be the equilibrium distribution of the unperturbed dynamics (drivings set to 0) subjected to the same constraints of the perturbed dynamics. For instance, if the perturbed dynamics preserves the kinetic energy thanks to isokinetic thermostats, the unperturbed dynamics must also preserve the kinetic energy, and $f_{0}$ must be invariant under the resulting (generally non-Hamiltonian) dynamics. In this case, analogous to the linear Formula (Equation 23), Equation (Equation 35) concerns the correlation function of the evolving observable of interest with the dissipative flux and such a correlation function is an average, computed with respect to the initial distribution $f_{0}$. The difference is that the dynamics followed by the observable are the exact perturbed dynamics and not the approximate equilibrium dynamics. However, the formalism can now be used more generally, without the need for $f_{0}$ to be the equilibrium distribution, or for the system of interest to be a particle system.

## 4. Exact Response of Stochastic Processes

In this section, we first adapt the exact response theory illustrated above to a generic abstract dynamical system perturbed by a stochastic, time-dependent, vector field, extending the approach for deterministic time-dependent perturbations developed in Ref. [15]. Then, we distinguish three cases: **(a)** The deterministic case in which a single realization of the perturbation is fixed (as in e.g., the Green–Kubo linear theory); thus, observables are only averaged with respect to the initial conditions of phase space trajectories; **(b)** The stochastic case in which the initial condition in the phase space is fixed. The realizations of the perturbation vary; hence, observables are averaged over the distribution of such realizations; **(c)** The stochastic case in which the phase space initial conditions and perturbation realizations vary, and observables are averaged over both.

### 4.1. Wiener Process Perturbation

Given a dynamical system $\dot{\Gamma }=G\left(\Gamma \right)$ on a phase space M, consider the following stochastic equation:(36)$$\dot{\Gamma }=\hat{G}\left(\Gamma ,t\right)=G\left(\Gamma \right)+F\hat{w}\left(t\right)\;,\;\mathrm{for}\;\;t\in \left[0,T\right]$$
where F is a constant that gives the strength of the perturbation, T>0 can be much larger than the scale of observation, as in [1], and w(t) is a Wiener process of the same dimension, as M. This can be represented by the Karhunen–Loève expansion, as follows: [34]:(37)$$\hat{w}\left(t\right)=\sum_{n=1}^{\infty }\frac{\sqrt{2T}}{\pi }\frac{Z_{n}}{\left(n-1/2\right)}\sin\left(\frac{(n-1/2)\pi }{T}\;t\right)=\sum_{n=-\infty }^{\infty }\alpha_{n}e^{i\beta_{n}t}$$
where the details of the process are determined by the vectors $Z_{n}=\left(Z_{1n},...,Z_{Nn}\right)$, or equivalently, $\alpha_{n}=\left(\alpha_{1n},\ldots ,\alpha_{Nn}\right)$, with:(38)$$\alpha_{kn}=\left\{\begin{array}{cc} \frac{1}{2i}\;\frac{\sqrt{2T}}{\pi }\;\frac{Z_{kn}}{c_{n}}\; & \mathrm{if}\;n>0\\ \frac{1}{2i}\;\frac{\sqrt{2T}}{\pi }\;\frac{Z_{-kn}}{c_{n}}\; & \mathrm{if}\;n<0 \end{array}\right.\;\;c_{n}=\left\{\begin{array}{cc} \; & n-\frac{1}{2}\;\mathrm{if}\;n>0,\\ \; & n+\frac{1}{2}\;\mathrm{if}\;n<0 \end{array}\right.\;\mathrm{and}\;\;\beta_{n}=\frac{c_{n}\pi }{T}$$
We recall that this expression holds for all perturbations in $L^{2}\left[0,T\right]$, for any T>0, so it is a very general result. As this is a zero mean process, it does not include a net force acting on the system. However, such a force, constituting a systematic action where the Wiener process fluctuates, can be included as a deterministic term in *G*.

Rather than directly computing the evolution of probability densities, we follow Ref. [15] in order to obtain a more flexible tool, and to preserve the notion of time reversal invariance. Therefore, we first eliminate the time dependence of the perturbation, and introduce two new variables (θ and ϕ) that evolve as follows:(39)$$\left\{\begin{array}{c} \dot{\theta }=\phi \\ \dot{\phi }=-\theta \end{array}\right.$$
These new variables live on an ellipse, whose axes depend on the initial independently chosen conditions, $\theta_{0}$ and $\phi_{0}$. Then, we can write the following:(40)$$F\hat{w}\left(t\right)=F\sum_{n=-\infty }^{\infty }\alpha_{n}e^{i\beta_{n}t}=\sum_{n=-\infty }^{\infty }\alpha_{n}w_{n}\left(\theta ,\phi \right)=w\left(\theta ,\phi \right)$$
where the deterministic parts of the perturbation are given by the functions $w_{n}\left(\theta ,\phi \right)$, and the stochastic parts by the coefficients $\alpha_{n}$. When F is given, we can write the following:(41)$$w_{n}\left(\theta ,\phi \right)=F^{1-\beta_{n}}{\left(\theta (t)-i\phi (t)\right)}^{\beta_{n}}\;\mathrm{with}\;n\in Z\;,\;\mathrm{and}\;\;\theta \left(t\right)-i\phi \left(t\right)=F\exp\left(it\right)$$
Alternatively, the initial condition determines F through the equality θ(t)−iϕ(t)=Fexp(it). The difference between the two situations depends on the initial distribution of the points (θ,ϕ), which we are now free to choose as the problem of interest requires.

In both cases, the explicit time dependence of the perturbation *w* is turned into an implicit dependence, mediated by θ and ϕ. The new dynamical system is as follows:(42)$$\dot{\tilde{\Gamma }}=\tilde{G}\left(\Gamma ,\theta ,\phi \right)=\left(\begin{array}{c} G(\Gamma )+w(\theta ,\phi )\\ \phi \\ -\theta \end{array}\right)$$
which is an autonomous system of differential equations, whose phases $\tilde{\Gamma }=\left(\Gamma ,\theta ,\phi \right)$ live in the new phase space
(43)$$\tilde{M}=M_{\Gamma }\times M_{\theta \phi }$$
where $M_{\Gamma }$ coincides with the original phase space of the unperturbed dynamics, M, while $M_{\theta \phi }\subset R^{2}$ is the space of the points (θ,ϕ), which depends on the problem at hand, and may be bounded or not. In particular, a given initial condition confines (θ,ϕ) to a circle of the given radius, F, for a harmonic oscillator in one dimension, whose energy is fixed by the initial conditions. The corresponding new phase space variation rate equals the old one:(44)$$\tilde{\Lambda }=\tilde{\Lambda }\left(\Gamma ,\theta ,\phi \right)=\Lambda \left(\Gamma \right)$$
because the new coordinates describe a Hamiltonian system. Introducing a new probability density, $\tilde{f_{0}}=\tilde{f_{0}}\left(\Gamma ,\theta ,\phi \right)$, a new dissipation function, ${\tilde{\Omega }}^{f_{0}}={\tilde{\Omega }}^{f_{0}}\left(\Gamma ,\theta ,\phi \right)$, can also be obtained, applying the general rule (Equation 29) to the quantities with a tilde. The density $\tilde{f_{0}}$ can be given in various guises. However, the distribution of θ and ϕ is not affected by the phase space coordinates Γ, and should not affect the distribution of Γ; therefore, it can be factorized as follows:(45)$${\tilde{f}}_{0}\left(\Gamma ,\theta ,\phi \right)=f_{0}\left(\Gamma \right)g_{0}\left(\theta ,\phi \right)$$
In the case that the realization of the perturbation is fixed, and its initial value is given by $w_{0}=w\left(\theta_{0},\phi_{0}\right)$, one could obtain the following:(46)$${\tilde{f}}_{0}\left(\Gamma ,\theta ,\phi \right)=f_{0}\left(\Gamma \right)\delta \left(\theta -\theta_{0}\right)\delta \left(\phi -\phi_{0}\right)$$
where $f_{0}$ is invariant under the unperturbed dynamics, as in the usual molecular dynamic applications of the exact response formula, and δ is the Dirac delta function. On the other hand, finite accuracy in setting the initial value of *w* may be described by replacing the δ-function with a smooth non-negative and normalized function, centered on $\theta_{0}$ and $\phi_{0}$, which vanishes in a suitably narrow or wide interval around 0. Physically, this is more meaningful; mathematically, it is useful since it preserves the differentiability of ${\tilde{f}}_{0}$. In this case, we would write the following:(47)$${\tilde{f}}_{0}\left(\Gamma ,\theta ,\phi \right)=f_{0}\left(\Gamma \right)\;g_{0}\left(\theta -\theta_{0}\right)\;g_{0}\left(\phi -\phi_{0}\right)$$
where the relation to $\theta_{0}$ and $\phi_{0}$ is stressed. If, on the other hand, the magnitude of the perturbation is fixed, one may take $M_{\theta \phi }=S^{F}$, the circle of radius F. Then, because this circle is traversed with uniform speed by the point (θ(t),ϕ(t)) and the origin of times is arbitrary, a natural form of the phase space distribution is as follows:(48)$$\tilde{f_{0}}\left(\Gamma ,\theta ,\phi \right)=\frac{1}{2\pi F}\;f_{0}\left(\Gamma \right)$$

Let us rewrite the equations of motion as follows:(49)$$\dot{\tilde{\Gamma }}\left(\Gamma ,\theta ,\phi \right)=\left(\begin{array}{c} {\dot{\Gamma }}_{1}\\ {\dot{\Gamma }}_{2}\\ \vdots \\ {\dot{\Gamma }}_{N}\\ \dot{\theta }\\ \dot{\phi } \end{array}\right)=\left(\begin{array}{c} G_{1}\left(\Gamma \right)+\sum_{n=-\infty }^{\infty }\alpha_{n1}w_{n}\left(\theta ,\phi \right)\\ G_{2}\left(\Gamma \right)+\sum_{n=-\infty }^{\infty }\alpha_{n2}w_{n}\left(\theta ,\phi \right)\\ \;\vdots \\ G_{N}\left(\Gamma \right)+\sum_{n=-\infty }^{\infty }\alpha_{nN}w_{n}\left(\theta ,\phi \right)\\ \;\;\phi \\ \;-\theta \end{array}\right)$$
where $(\Gamma_{1},\ldots ,\Gamma_{N})=\Gamma \in M$, and $(G_{1},...,G_{N})=G$. Given the initial distribution, the new dissipation function is derived as follows:(50)$$\begin{array}{ccc} & & \;{\tilde{\Omega }}_{{\tilde{f}}_{0}}\left(\Gamma ,\theta ,\phi \right)=-\tilde{\Lambda }\left(\Gamma ,\theta ,\phi \right)-\frac{d}{dt}\ln\tilde{f_{0}}\left(\Gamma ,\theta ,\phi \right)\\ & & \;=-\Lambda \left(\Gamma \right)-\frac{1}{f_{0}\left(\Gamma \right)g_{0}\left(\theta ,\phi \right)}[g_{0}\left(\theta ,\phi \right)\left(G(\Gamma )+w(\theta ,\phi )\right)\cdot \nabla_{\Gamma }f_{0}\left(\Gamma \right)+\;\\ & & \;+f_{0}\left(\Gamma \right)\left(\phi \;\frac{\partial g_{0}}{\partial \theta }\left(\theta ,\phi \right)-\theta \;\frac{\partial g_{0}}{\partial \phi }\left(\theta ,\phi \right)\right)]\\ & & \;=-\Lambda \left(\Gamma \right)-\frac{1}{f_{0}\left(\Gamma \right)}\sum_{k=1}^{N}\left[G_{k}\left(\Gamma \right)+\sum_{n=-\infty }^{\infty }\alpha_{nk}w_{n}\left(\theta ,\phi \right)\right]\frac{\partial f_{0}}{\partial \Gamma_{k}}\left(\Gamma \right)\\ & & \;-\frac{1}{g_{0}\left(\theta ,\phi \right)}\left(\phi \;\frac{\partial g_{0}}{\partial \theta }\left(\theta ,\phi \right)-\theta \;\frac{\partial g_{0}}{\partial \phi }\left(\theta ,\phi \right)\right)\\ & & \;=\Omega_{f_{0}}\left(\Gamma \right)-\frac{1}{f_{0}\left(\Gamma \right)}\sum_{n=-\infty }^{\infty }w_{n}\left(\theta ,\phi \right)\sum_{k=1}^{N}\alpha_{nk}\frac{\partial f_{0}}{\partial \Gamma_{k}}\left(\Gamma \right)-\frac{1}{g_{0}\left(\theta ,\phi \right)}\left(\phi \;\frac{\partial g_{0}}{\partial \theta }\left(\theta ,\phi \right)-\theta \;\frac{\partial g_{0}}{\partial \phi }\left(\theta ,\phi \right)\right) \end{array}$$
Concerning the observables $\tilde{O}$ defined on $\tilde{M}$, we only need those that do not depend on θ and ϕ, i.e., such that $\tilde{O}\left(\Gamma ,\theta ,\phi \right)=O\left(\Gamma \right)$, where O is one observable defined on M. Nevertheless, the correlation of ${\tilde{\Omega }}_{{\tilde{f}}_{0}}$ with the time evolution of one such observable must be written as follows:(51)$$\begin{array}{ccc} {\langle {\tilde{\Omega }}_{{\tilde{f}}_{0}}\left(\tilde{O}\circ {\tilde{S}}^{s}\right)\rangle }_{{\tilde{f}}_{0}} & = & \int_{M\times M_{\theta \phi }}{\tilde{\Omega }}_{{\tilde{f}}_{0}}\left(\Gamma ,\theta ,\phi \right)\tilde{O}\left({\tilde{S}}^{s}\left(\Gamma ,\theta ,\phi \right)\right)f_{0}\left(\Gamma \right)g_{0}\left(\theta ,\phi \right)d\Gamma d\theta d\phi \end{array}$$
(52)$$\begin{array}{ccc} & = & \int_{M_{\theta \phi }}d\theta d\phi \;g_{0}\left(\theta ,\phi \right)\int_{M}\;d\Gamma \;{\tilde{\Omega }}_{{\tilde{f}}_{0}}\left(\Gamma ,\theta ,\phi \right)\tilde{O}\left({\tilde{S}}^{s}\;\tilde{\Gamma }\right)f_{0}\left(\Gamma \right) \end{array}$$
because $\tilde{O}$ does not depend on the last two components of $\tilde{\Gamma }$ and can be replaced by O; however, the time evolution of $\tilde{\Gamma }$ and, hence, of the phases Γ∈M, depends on Γ, θ, and ϕ.

In the case where the time-dependent perturbation is fixed at time 0, i.e., the distribution $g_{0}$ is a delta function centered on the required initial values of θ and ϕ, the dependence of ${\tilde{S}}^{s}\;\tilde{\Gamma }$ on such initial conditions can be absorbed in the deterministic part of the dynamics, and the different contributions to the above expressions take the following form. The third addend of ${\tilde{\Omega }}_{{\tilde{f}}_{0}}$ does not depend on Γ, and $\tilde{O}\left({\tilde{S}}^{s}\;\tilde{\Gamma }\right)=O\left(S^{s}\Gamma \right)$; therefore, it yields:(53)$$\begin{array}{ccc} & & \;-\int_{M_{\theta \phi }}d\theta d\phi \;g_{0}\left(\theta ,\phi \right)\frac{1}{g_{0}\left(\theta ,\phi \right)}\left(\phi \;\frac{\partial g_{0}}{\partial \theta }\left(\theta ,\phi \right)-\theta \;\frac{\partial g_{0}}{\partial \phi }\left(\theta ,\phi \right)\right)\int_{M}\;d\Gamma \;\tilde{O}\left({\tilde{S}}^{s}\;\tilde{\Gamma }\right)f_{0}\left(\Gamma \right) \end{array}$$
(54)$$\begin{array}{ccc} & & =-{\langle O\circ S^{s}\Gamma \rangle }_{0}\;\int_{M_{\theta \phi }}d\theta d\phi \;\left(\phi \;\frac{\partial g_{0}}{\partial \theta }\left(\theta ,\phi \right)-\theta \;\frac{\partial g_{0}}{\partial \phi }\left(\theta ,\phi \right)\right) \end{array}$$
Because variables θ and ϕ are only introduced as auxiliary quantities and have no effect on the evolution in M and the evolution of observables, their distribution $g_{0}$ can be chosen quite freely. Moreover, their roles are fully interchangeable, so we can take distributions such as $g_{0}\left(\theta ,\phi \right)=g_{0}\left(\phi ,\theta \right)$, which make the integral in expression (54) vanish.

The first addend in the expression of ${\tilde{\Omega }}_{{\tilde{f}}_{0}}$ yields:(55)$$\begin{array}{ccc} & & \;\int_{M_{\theta \phi }}d\theta d\phi \;g_{0}\left(\theta ,\phi \right)\int_{M}\;d\Gamma \;\Omega_{f_{0}}\left(\Gamma \right)O\left(S^{s}\Gamma \right)f_{0}\left(\Gamma \right) \end{array}$$
(56)$$\begin{array}{ccc} & & \;=\int_{M}\;d\Gamma \;\Omega_{f_{0}}\left(\Gamma \right)O\left(S^{s}\Gamma \right)f_{0}\left(\Gamma \right)={\langle \Omega_{f_{0}}\left(O\circ S^{s}\Gamma \right)\rangle }_{0} \end{array}$$
because $g_{0}$ is normalized. The remaining term, concerning the time-dependent perturbation, finally yields:(57)$$\begin{array}{c} -\int_{M_{\theta \phi }}d\theta d\phi \;g_{0}\left(\theta ,\phi \right)\int_{M}\;d\Gamma \;\frac{1}{f_{0}\left(\Gamma \right)}\sum_{n=-\infty }^{\infty }w_{n}\left(\theta ,\phi \right)\sum_{k=1}^{N}\alpha_{nk}\frac{\partial f_{0}}{\partial \Gamma_{k}}\left(\Gamma \right)O\left(S^{s}\Gamma \right)f_{0}\left(\Gamma \right) \end{array}$$
(58)$$\begin{array}{c} -\sum_{n=-\infty }^{\infty }\sum_{k=1}^{N}\alpha_{nk}\int_{M_{\theta \phi }}d\theta d\phi \;w_{n}\left(\theta ,\phi \right)g_{0}\left(\theta ,\phi \right)\int_{M}\;d\Gamma \;\frac{\partial f_{0}}{\partial \Gamma_{k}}\left(\Gamma \right)O\left(S^{s}\Gamma \right) \end{array}$$
At this point, it remains to decide how to deal with the different realizations of the perturbation, namely how to treat the stochastic vectors $\alpha_{n}$.

### 4.2. Single Perturbation Realization

Here, we treat our system as deterministic, subjected to the time-dependent perturbation that corresponds to a single realization of the stochastic perturbation. Let us label by (j) this realization. In this case, we can write:(59)$${\dot{\tilde{\Gamma }}}^{\left(j\right)}={\tilde{G}}^{\left(j\right)}\left(\Gamma ,\theta ,\phi \right)=G\left(\Gamma \right)+\sum_{n=-\infty }^{\infty }\alpha_{n}^{\left(j\right)}w_{n}\left(\theta ,\phi \right)$$
where the expansion coefficients, denoted by $\alpha_{n}^{\left(j\right)}=\left(\alpha_{n1}^{\left(j\right)},...\alpha_{nN}^{\left(j\right)}\right)$, characterize the chosen perturbation (j), and the initial condition and the magnitude of the perturbation are fixed by θ and ϕ. The phase space expansion rate is given by
(60)$$\tilde{\Lambda }\left(\Gamma ,\theta ,\phi \right)=\nabla_{\tilde{\Gamma }}\cdot {\tilde{G}}^{\left(j\right)}=\nabla_{\Gamma }\cdot G=\Lambda \left(\Gamma \right)$$
where Λ denotes the unperturbed phase space volume variation rate. The new dissipation function is expressed by
(61)$$\begin{array}{ccc} & & \;{\tilde{\Omega }}_{{\tilde{f}}_{0}}^{\left(j\right)}\left(\Gamma ,\theta ,\phi \right)=\\ & & \Omega_{f_{0}}\left(\Gamma \right)-\frac{1}{f_{0}\left(\Gamma \right)}\sum_{n=-\infty }^{\infty }w_{n}\left(\theta ,\phi \right)\sum_{k=1}^{N}\alpha_{nk}^{\left(j\right)}\;\frac{\partial f_{0}}{\partial \Gamma_{k}}\left(\Gamma \right)-\frac{1}{g_{0}\left(\theta ,\phi \right)}\left(\phi \;\frac{\partial g_{0}}{\partial \theta }\left(\theta ,\phi \right)-\theta \;\frac{\partial g_{0}}{\partial \phi }\left(\theta ,\phi \right)\right) \end{array}$$
where $g_{0}$ is the distribution of the initial condition $\left(\theta_{0},\phi_{0}\right)$ of the auxiliary variables. Choosing $g_{0}$ as in the previous section, the last addend of ${\tilde{\Omega }}_{{\tilde{f}}_{0}}^{\left(j\right)}$ does not affect the response of observables. Therefore, given an observable O (which depends on Γ only), and denoting by $S^{{\left(j\right)}^{s}}$ the evolution operator of the perturbed dynamics in M, we can write the following:(62)$$\begin{array}{ccc} {\langle O\rangle }_{{\tilde{f}}_{t}} & = & {\langle O\rangle }_{0}+\int_{0}^{t}{\langle \left(O\circ S^{{\left(j\right)}^{s}}\right)\;{\tilde{\Omega }}_{{\tilde{f}}_{0}}^{\left(j\right)}\rangle }_{0}\;ds \end{array}$$
(63)$$\begin{array}{ccc} & = & {\langle O\rangle }_{0}+\int_{0}^{t}{\langle \left(O\circ S^{{\left(j\right)}^{s}}\right)\;\Omega_{f_{0}}\rangle }_{0}\;ds\\ & & -\int_{0}^{t}ds\sum_{n=-\infty }^{\infty }\sum_{k=1}^{N}\alpha_{nk}^{\left(j\right)}\int_{M_{\theta \phi }}d\theta d\phi \;w_{n}\left(\theta ,\phi \right)g_{0}\left(\theta ,\phi \right)\int_{M}\;d\Gamma \;\frac{\partial f_{0}}{\partial \Gamma_{k}}\left(\Gamma \right)O\left(S^{{\left(j\right)}^{s}}\Gamma \right) \end{array}$$
(64)$$\begin{array}{ccc} & = & {\langle O\rangle }_{0}+\int_{0}^{t}{\langle \left(O\circ S^{{\left(j\right)}^{s}}\right)\;\Omega_{f_{0}}\rangle }_{0}\;ds\\ & & -F\sum_{n=-\infty }^{\infty }\sum_{k=1}^{N}\alpha_{nk}^{\left(j\right)}\int_{M_{\theta \phi }}d\theta d\phi \;F^{-\beta_{n}}{\left(\theta -i\phi \right)}^{\beta_{n}}g_{0}\left(\theta ,\phi \right)\int_{0}^{t}ds\int_{M}\;d\Gamma \;\frac{\partial f_{0}}{\partial \Gamma_{k}}\left(\Gamma \right)O\left(S^{{\left(j\right)}^{s}}\Gamma \right) \end{array}$$
where $F^{-\beta_{n}}{\left(\theta -i\phi \right)}^{\beta_{n}}$ does not depend on F, cf. Equation (Equation 41). Here, the integral concerning the auxiliary variables give the mean of $w_{n}$ with respect to the initial distribution of (θ,ϕ). If this is fixed, then this integral simply gives
(65)$$w_{n}\left(\theta_{0},\phi_{0}\right)=FK_{\theta_{0}\phi_{0}}^{(n\pm 1/2)\pi /T}$$
with $K_{\theta_{0}\phi_{0}}$ being a constant that depends on the initial condition, raised to the power of (n−1/2)π/T if n>0 and to the power of (n+1/2)π/T if n<0. But the situation is analogous if the initial values of θ and ϕ are distributed with any other density $g_{0}$. Naturally, letting F vanish makes the stochastic contribution vanish as well, thus returning to the response formula for time-independent perturbations. So far, we have considered the coefficients $\alpha_{nk}^{\left(j\right)}$ as fixed, which makes the dynamics look deterministic, and only averages over the initial conditions in the phase space.

### 4.3. Stochastic Coefficients with Fixed Initial Condition

Let us now average over the distribution of the stochastic coefficients, considering the first and second cumulants of the fluctuations generated by the stochastic perturbation, in relation to the response due to the deterministic term *G*. The higher-order cumulants vanish since the stochastic coefficients are Gaussian random variables. Keeping the initial condition $\left(\Gamma ,\theta ,\phi \right)\in \hat{M}$ fixed, we denote by ${\langle \;\cdot \;\rangle }^{\left(st\right)}$ the averages made with respect to all realizations of the stochastic perturbation. Consider the equation of motion (Equation 42) and its first cumulant:(66)$${\langle \dot{\tilde{\Gamma }}\rangle }^{\left(st\right)}=\tilde{G}+\sum_{n=-\infty }^{\infty }{\langle \alpha_{n}\rangle }^{\left(st\right)}w_{n}=\tilde{G}$$
As anticipated, the first cumulant equals the cumulant of the deterministic part of the equation of motion. Moreover, integrating in time, we obtain the following:(67)$${\langle \tilde{\Gamma }\left(t\right)-\tilde{\Gamma }\left(0\right)\rangle }^{\left(st\right)}=\int_{0}^{t}ds\;{\langle \dot{\tilde{\Gamma }}\left(s\right)\rangle }^{\left(st\right)}=\int_{0}^{t}ds\;\tilde{G}\left(\Gamma \left(s\right)\right)=\left(\begin{array}{c} \Gamma (t)\\ \theta (t)\\ \theta (t) \end{array}\right)-\left(\begin{array}{c} \Gamma_{0}\\ \theta_{0}\\ \theta_{0} \end{array}\right)$$
For the second cumulant, let us consider, for instance, the component $\Gamma_{l}$ of Γ. We find the following:(68)$${\langle {\dot{\Gamma }}_{l}^{2}\rangle }^{\left(st\right)}-{\langle {\dot{\Gamma }}_{l}\rangle }^{{\left(st\right)}^{2}}={\langle {\dot{\Gamma }}_{l}^{2}\rangle }^{\left(st\right)}-G_{l}^{2}$$
Then, let us compute the following:(69)$$\begin{array}{ccc} & & \;{\langle {\dot{\Gamma }}_{l}\left(t\right){\dot{\Gamma }}_{l}\left(t^{\prime }\right)\rangle }^{\left(st\right)}={\langle \left[G_{l}\left(\Gamma \left(t\right)\right)+\sum_{n=-\infty }^{\infty }\alpha_{nl}w_{n}\left(\theta \left(t\right),\phi \left(t\right)\right)\right]\left[G_{l}\left(\Gamma \left(t’\right)\right)+\sum_{n=-\infty }^{\infty }\alpha_{nl}w_{n}\left(\theta \left(t’\right),\phi \left(t’\right)\right)\right]\rangle }^{\left(st\right)}\\ & & ={\langle G_{l}\left(\Gamma \left(t\right)\right)G_{l}\left(\Gamma \left(t^{\prime }\right)\right)\rangle }^{\left(st\right)}+G_{l}\left(\Gamma \left(t\right)\right)\sum_{n=-\infty }^{\infty }{\langle \alpha_{nl}\rangle }^{\left(st\right)}w_{n}\left(\theta \left(t^{\prime }\right),\phi \left(t^{\prime }\right)\right)\\ & & \;+\;G_{l}\left(\Gamma \left(t^{\prime }\right)\right)\sum_{n=-\infty }^{\infty }{\langle \alpha_{nl}\rangle }^{\left(st\right)}w_{n}\left(\theta \left(t\right),\phi \left(t\right)\right)\\ & & \;+\sum_{n=-\infty }^{\infty }\sum_{k=-\infty }^{\infty }{\langle \alpha_{nl}\alpha_{kl}\rangle }^{\left(st\right)}w_{n}\left(\theta (t),\phi (t)\right)w_{k}\left(\theta (t’),\phi (t’)\right) \end{array}$$
(70)$$\begin{array}{ccc} & & =G_{l}\left(\Gamma (t)\right)G_{l}\left(\Gamma (t^{\prime })\right)+\sum_{n=-\infty }^{\infty }{\langle \alpha_{nl}^{2}\rangle }^{\left(st\right)}w_{n}\left(\theta (t),\phi (t)\right)w_{n}\left(\theta (t’),\phi (t’)\right) \end{array}$$
Indeed, recalling the fact that the random coefficients $Z_{k}$ are delta-correlated, we have
(71)$${\langle \alpha_{kl}\alpha_{n-k,l}\rangle }^{\left(st\right)}=\left\{\begin{array}{cc} \frac{1}{4i^{2}}\frac{T}{\pi^{2}}\frac{{\langle Z_{kl}Z_{n-k,l}\rangle }^{\left(st\right)}}{c_{k}c_{n-k}}=-\frac{T}{{\left(2\pi \right)}^{2}}\frac{\delta_{k,n-k}}{c_{k}c_{n-k}} & \mathrm{if}\;k,n>0\\ \frac{1}{4i^{2}}\frac{2T}{\pi^{2}}\frac{{\langle Z_{-kl}Z_{-n+k,l}\rangle }^{\left(st\right)}}{c_{k}c_{n-k}}=-\frac{T}{{\left(2\pi \right)}^{2}}\frac{\delta_{-k,-n+k}}{c_{k}c_{n-k}} & \mathrm{if}\;k,n<0 \end{array}\right.$$
so, $\delta_{k,n-k}=1$ and $\delta_{-k,-n+k}=1$ if and only if n=2k. Consequently,
(72)$${\langle \alpha_{kl}^{2}\rangle }^{\left(st\right)}=-\frac{T}{{\left(2\pi \right)}^{2}}\frac{1}{c_{k}^{2}}.$$
Now, setting $t=t^{\prime }$, we obtain the following:(73)$${\langle {\dot{\Gamma }}_{l}{\left(t\right)}^{2}\rangle }^{\left(st\right)}=G_{l}{\left(\Gamma \left(t\right)\right)}^{2}+\sum_{n=-\infty }^{\infty }{\langle \alpha_{nl}^{2}\rangle }^{\left(st\right)}w_{n}{\left(\theta (t),\phi (t)\right)}^{2}$$
and, as a result, the second cumulant takes the following simple form:(74)$$\begin{array}{ccc} {\langle {\dot{\Gamma }}_{l}^{2}\left(t\right)\rangle }^{\left(st\right)}-{\langle {\dot{\Gamma }}_{l}\left(t\right)\rangle }^{{\left(st\right)}^{2}} & = & G_{l}{\left(\Gamma (t)\right)}^{2}+\sum_{k=-\infty }^{\infty }{\langle \alpha_{kl}^{2}\rangle }^{\left(st\right)}w_{k}{\left(\theta (t),\phi (t)\right)}^{2}-G_{l}{\left(\Gamma (t)\right)}^{2} \end{array}$$
(75)$$\begin{array}{ccc} & = & \sum_{k=-\infty }^{\infty }{\langle \alpha_{k}^{2}\rangle }^{\left(st\right)}w_{k}{\left(\theta (t),\phi (t)\right)}^{2} \end{array}$$
Integrating in time expression (70), we obtain
(76)$$\begin{array}{ccc} & & \;{\langle \int_{0}^{t}ds\;{\dot{\Gamma }}_{l}\left(s\right)\int_{0}^{t^{\prime }}ds^{\prime }\;{\dot{\Gamma }}_{l}\left(s’\right)\rangle }^{\left(st\right)}={\langle \left(\Gamma_{l}\left(t\right)-\Gamma_{l}\left(0\right)\right)\left(\Gamma_{l}\left(t’\right)-\Gamma_{l}\left(0\right)\right)\rangle }^{\left(st\right)}\\ & & =\int_{0}^{t}ds\int_{0}^{t^{\prime }}ds^{\prime }\left[G_{l}\left(\Gamma \left(s\right)\right)G_{l}\left(\Gamma \left(s^{\prime }\right)\right)+\sum_{k=-\infty }^{\infty }{\langle \alpha_{kl}^{2}\rangle }^{\left(st\right)}w_{k}\left(\theta (s),\phi (s)\right)w_{k}\left(\theta (s’),\phi (s’)\right)\right] \end{array}$$
and note that Equations (Equation 40) and (Equation 41) imply:(77)$$\frac{d}{ds}w_{n}\left(\theta (s),\phi (s)\right)=i\beta_{n}w_{n}\left(\theta (s),\phi (s)\right)$$
so that we can write
(78)$$\int_{0}^{t}w_{n}\left(\theta (s),\phi (s)\right)ds=\frac{1}{i\beta_{n}}\;\left[w_{n}\left(\theta (t),\phi (t)\right)-w_{n}\left(\theta_{0},\phi_{0}\right)\right]$$
Then, we obtain the following:(79)$$\begin{array}{ccc} & & \;{\langle \left(\Gamma_{l}\left(t\right)-\Gamma_{l}\left(0\right)\right)\left(\Gamma_{l}\left(t^{\prime }\right)-\Gamma_{l}\left(0\right)\right)\rangle }^{\left(st\right)}=\\ & & \;\int_{0}^{t}ds\int_{0}^{t^{\prime }}ds^{\prime }G_{l}\left(\Gamma (s)\right)G_{l}\left(\Gamma (s^{\prime })\right)+\sum_{k=-\infty }^{\infty }\frac{{\langle \alpha_{kl}^{2}\rangle }^{\left(st\right)}}{{\left(i\beta_{k}\right)}^{2}}\left[w_{k}\left(\theta (t),\phi (t)\right)-w_{k}\left(\theta_{0},\phi_{0}\right)\right]\\ & & \;\times \left[w_{k}\left(\theta \left(t^{\prime }\right),\phi \left(t^{\prime }\right)\right)-w_{k}\left(\theta_{0},\phi_{0}\right)\right] \end{array}$$
Consequently, observing that
(80)$$\begin{array}{ccc} & & \;{\langle \left(\Gamma_{l}\left(t\right)-\Gamma \left(0\right)\right)\left(\Gamma_{l}\left(t^{\prime }\right)-\Gamma_{l}\left(0\right)\right)\rangle }^{\left(st\right)}-{\langle \Gamma_{l}\left(t\right)-\Gamma_{l}\left(0\right)\rangle }^{\left(st\right)}{\langle \Gamma_{l}\left(t^{\prime }\right)-\Gamma_{l}\left(0\right)\rangle }^{\left(st\right)} \end{array}$$
(81)$$\begin{array}{ccc} & & \;={\langle \Gamma_{l}\left(t\right)\Gamma_{l}\left(t^{\prime }\right)\rangle }^{\left(st\right)}-{\langle \Gamma_{l}\left(t\right)\rangle }^{\left(st\right)}{\langle \Gamma_{l}\left(t^{\prime }\right)\rangle }^{\left(st\right)} \end{array}$$
the autocorrelation function is expressed by the following:(82)$$\begin{array}{cc} \; & \;{\langle \Gamma_{l}\left(t\right)\Gamma_{l}\left(t^{\prime }\right)\rangle }^{\left(st\right)}-{\langle \Gamma_{l}\left(t\right)\rangle }^{\left(st\right)}{\langle \Gamma_{l}\left(t^{\prime }\right)\rangle }^{\left(st\right)}=\\ \; & \sum_{k=-\infty }^{\infty }\frac{{\langle \alpha_{kl}^{2}\rangle }^{\left(st\right)}}{{\left(i\beta_{k}\right)}^{2}}\left[w_{k}\left(\theta (t),\phi (t)\right)-w_{k}\left(\theta_{0},\phi_{0}\right)\right]\left[w_{k}\left(\theta (t’),\phi (t’)\right)-w_{k}\left(\theta_{0},\phi_{0}\right)\right]\\ \; & \;+\int_{0}^{t}ds\int_{0}^{t^{\prime }}ds^{\prime }\left[G_{l}\left(\Gamma \left(s\right)\right)G_{l}\left(\Gamma \left(s^{\prime }\right)\right)-G_{l}\left(\Gamma \left(s\right)\right)G_{l}\left(\Gamma \left(s^{\prime }\right)\right)\right]\\ \; & =\sum_{k=-\infty }^{\infty }\frac{{\langle \alpha_{kl}^{2}\rangle }^{\left(st\right)}}{{\left(i\beta_{k}\right)}^{2}}\left[w_{k}\left(\theta (t),\phi (t)\right)-w_{k}\left(\theta_{0},\phi_{0}\right)\right]\left[w_{k}\left(\theta (t’),\phi (t’)\right)-w_{k}\left(\theta_{0},\phi_{0}\right)\right] \end{array}$$
and $t=t^{\prime }$ yields:(83)$$\begin{array}{cc} {\langle \Gamma {\left(t\right)}^{2}\rangle }^{\left(st\right)}-{\langle \Gamma (t)\rangle }^{{\left(st\right)}^{2}} & =\sum_{k=-\infty }^{\infty }\frac{{\langle \alpha_{kl}^{2}\rangle }^{\left(st\right)}}{{\left(i\beta_{k}\right)}^{2}}{\left[w_{k}\left(\theta (t),\phi (t)\right)-w_{k}\left(\theta_{0},\phi_{0}\right)\right]}^{2} \end{array}$$
For the dissipation function, the first cumulant is trivial since the stochastic coefficients have a zero mean:(84)$${\langle {\tilde{\Omega }}_{f_{0}}\rangle }^{\left(st\right)}={\langle \Omega_{f_{0}}\rangle }^{\left(st\right)}$$
For the second cumulant, we have the following:
(85)Ω˜f0−Ω˜f0(st)2(st)=Ω˜f02(st)−Ω˜f0(st)2(86)=Ωf0w∂lnf0∂ΓΩf0w∂lnf0∂Γ(st)−Ωf0(st)2(87)=Ωf02(st)−Ωf02+∑k=−∞∞αk2(st)wk2∂lnf0∂Γ2
thanks to the statistical properties of stochastic coefficients.

### 4.4. Averaging over Initial Conditions and Stochastic Coefficients

Given a single realization of the stochastic perturbation, the response formula for an observable O takes the following form: (88)$${\langle O\rangle }_{t}={\langle O\rangle }_{0}+\int_{0}^{t}{\langle \left(O\circ S^{{\left(j\right)}^{s}}\right)\Omega^{f_{0}}\rangle }_{0}\;ds$$
If we further average over the realizations of the stochastic process, we use the following notation:(89)$${\langle {\langle O\rangle }_{t}\rangle }^{\left(st\right)}$$
and we can write: (90)$$\begin{array}{ccc} {\langle {\langle O\rangle }_{{\tilde{f}}_{t}}\rangle }^{\left(st\right)} & = & {\langle {\langle O\rangle }_{{\tilde{f}}_{0}}\rangle }^{\left(st\right)}+\int_{0}^{t}{\langle {\langle \left(O\circ S^{{\left(j\right)}^{s}}\right){\tilde{\Omega }}_{f_{0}}\rangle }_{{\tilde{f}}_{0}}\rangle }^{\left(st\right)}\;ds\\ & = & {\langle O\rangle }_{{\tilde{f}}_{0}}+\int_{0}^{t}{\langle {\langle \left(O\circ S^{{\left(j\right)}^{s}}\right)\left[\Omega_{f_{0}}-w\frac{\partial \ln f_{0}\left(\Gamma \right)}{\partial \Gamma }\right]\rangle }_{{\tilde{f}}_{0}}^{\left(st\right)}\rangle }^{\left(st\right)}\;ds\\ & = & {\langle O\rangle }_{t}-{\langle \sum_{n=-\infty }^{+\infty }\alpha_{n}\int_{0}^{t}ds{\langle \left(O\circ S^{s}\right)w_{n}\frac{\partial \ln f_{0}\left(\Gamma \right)}{\partial \Gamma }\rangle }_{{\tilde{f}}_{0}}\rangle }^{\left(st\right)} \end{array}$$

## 5. Stochastic Single Frequency Periodic Force

To illustrate the theory developed above, let us consider a simple example, which can be treated analytically. More complex situations can be dealt with by performing numerical integration. Let us consider a harmonic oscillator in one dimension, which is perturbed with a time-dependent force. We will compare the linear and the exact response. Let $H_{0}=\frac{p^{2}}{2}+\frac{\omega^{2}q^{2}}{2}$ be the unperturbed Hamiltonian, and consider the following perturbation:(91)$$H_{p}=-w\left(t\right)q=-\epsilon Z\sin\left(\gamma t\right)q$$
where ϵ is a pure number, and *Z* has the dimension of force, which could be deterministic as a special case but is stochastic in general. Then, the perturbed energy is given by
(92)$$H\left(q,p,t\right)=H_{0}+H_{p}=\frac{p^{2}+\omega^{2}q^{2}}{2}-w\left(t\right)q$$
and the equations of motion are expressed by the following:(93)$$G\left(q,p,t\right)=\left(\begin{array}{c} \dot{q}\\ \dot{p} \end{array}\right)=\left(\begin{array}{c} p\\ w\left(t\right)-\omega^{2}q \end{array}\right)$$
We make the system autonomous, introducing two new variables, θ and ϕ, so that the dynamics are given by the following:(94)$$\tilde{G}\left(\Gamma ,\theta ,\phi \right)=\left(\begin{array}{c} \dot{q}\\ \dot{p}\\ \dot{\theta }\\ \dot{\phi } \end{array}\right)=\left(\begin{array}{c} p\\ w\left(\theta ,\phi \right)-\omega^{2}q\\ \phi \\ -\gamma^{2}\theta \end{array}\right)$$
and the perturbation can be written as follows:(95)w(t)=w(θ(t),ϕ(t))=−ϵZϕ(t)
The motion is then given by
(96)$${\tilde{S}}_{\omega }^{t}\left(\begin{array}{c} q\\ p\\ \theta \\ \phi \end{array}\right)=\left(\begin{array}{c} C_{1}\cos\left(\omega t\right)+\frac{C_{2}}{\omega }\sin\left(\omega t\right)+\frac{\epsilon Z}{\left(\omega^{2}-\gamma^{2}\right)}\left[\theta_{0}\sin\left(\gamma t\right)-\frac{\phi_{0}}{\gamma }\cos\left(\gamma t\right)\right]\\ -\omega C_{1}\sin\left(\omega t\right)+C_{2}\cos\left(\omega t\right)+\frac{\epsilon Z}{\left(\omega^{2}-\gamma^{2}\right)}\left[\theta_{0}\gamma \cos\left(\gamma t\right)+\phi_{0}\sin\left(\gamma t\right)\right]\\ \theta_{0}\cos\left(\gamma t\right)+\frac{\phi_{0}}{\gamma }\sin\left(\gamma t\right)\\ -\gamma \theta_{0}\sin\left(\gamma t\right)+\phi_{0}\cos\left(\gamma t\right) \end{array}\right)$$
where
(97)$$C_{1}=\left(q_{0}+\epsilon Z\frac{\phi_{0}}{\gamma (\omega^{2}-\gamma^{2})}\right)\;,\;C_{2}=\left(p_{0}-\epsilon Z\frac{\gamma \theta_{0}}{\left(\omega^{2}-\gamma^{2}\right)}\right)$$
Let us compare the linear and exact responses of our system to the perturbation. First, suppose that $\tilde{f_{0}}$ takes the following form:(98)$$\tilde{f_{0}}\left(\Gamma ,\theta ,\phi \right)=\frac{f_{0}\left(\Gamma \right)}{\sigma }=\frac{1}{\sigma }\frac{e^{-\beta H_{0}}}{Z_{0}},$$
where σ is the circumference of the variables θ and ϕ, whose radius depends on the initial condition $F=\sqrt{\theta_{0}^{2}+\phi_{0}^{2}}$. Then,
(99)$$\tilde{\Lambda }\left(\Gamma ,\theta ,\phi \right)=\nabla_{\tilde{\Gamma }}\cdot \tilde{G}\left(\Gamma ,\theta ,\phi \right)=\frac{\partial }{\partial q}\dot{q}+\frac{\partial }{\partial p}\dot{p}+\frac{\partial }{\partial \theta }\dot{\theta }+\frac{\partial }{\partial \phi }\dot{\phi }=0,$$
(100)$$\begin{array}{cc} {\tilde{\Omega }}^{f_{0}}\left(\Gamma ,\theta ,\phi \right) & =-\tilde{\Lambda }\left(\Gamma ,\theta ,\phi \right)-\tilde{G}\left(\Gamma ,\theta ,\phi \right)\cdot \nabla_{\tilde{\Gamma }}\ln{\tilde{f}}_{0}\left(\Gamma ,\theta ,\phi \right)\\ \; & =-\left(p,w\left(\theta ,\phi \right)-\omega^{2}q,\phi ,-\gamma^{2}\theta \right)\cdot \left(\begin{array}{cc} \; & -\beta \omega^{2}q\\ \; & -\beta p\\ \; & 0\\ \; & 0 \end{array}\right)\\ \; & =\beta \omega^{2}pq+\beta w\left(\theta ,\phi \right)p-\beta \omega^{2}qp=\beta w\left(\theta ,\phi \right)p \end{array}$$
The exact response theory yields the following:(101)$${\langle O\rangle }_{{\tilde{f}}_{t}}^{\left(st\right)}={\langle O\rangle }_{{\tilde{f}}_{0}}^{\left(st\right)}+\int_{0}^{t}{\langle \left(O\circ S_{w}^{s}\right)\cdot {\tilde{\Omega }}^{f_{0}}\rangle }_{{\tilde{f}}_{0}}^{\left(st\right)}\;ds,$$
where $S_{w}^{s}$ denotes the perturbed dynamics and the ensemble average is taken over the augmented phase space. The linear response theory yields the following:(102)$${\langle O\rangle }_{f_{t}}^{\left(st\right)}={\langle O\rangle }_{f_{0}}^{\left(st\right)}+\beta \int_{0}^{t}F{\langle \left(O\circ S_{0}^{s}\right)\cdot \dot{A}\rangle }_{f_{0}}^{\left(st\right)}\;ds$$
where $S_{0}^{s}$ denotes the unperturbed dynamics.

Let us consider, for instance, O=q. To proceed, we need some preliminary results.
(103)$$\begin{array}{cc} {\langle q\rangle }_{f_{0}}^{\left(st\right)} & =\;\int \int \;dqdp\frac{e^{-\beta H_{0}}}{Z_{0}}q=\frac{\int_{-\infty }^{\infty }e^{-\frac{\beta p^{2}}{2}}dp\int_{-\infty }^{\infty }qe^{-\frac{\beta \omega^{2}q^{2}}{2}}dq}{\;\int \int \;e^{-\beta H_{0}}dqdp}\\ \; & =\frac{1}{\int_{-\infty }^{\infty }e^{-\frac{\beta \omega^{2}q^{2}}{2}}dq}\frac{-e^{-\frac{\beta \omega^{2}q^{2}}{2}}}{\beta \omega^{2}}{|}_{-\infty }^{\infty }=\frac{1}{\sqrt{2\pi /\beta \omega^{2}}}\frac{-e^{-\frac{\beta \omega^{2}q^{2}}{2}}}{\beta \omega^{2}}{|}_{-\infty }^{\infty }=0, \end{array}$$
(104)$$\begin{array}{cc} {\langle p\rangle }_{f_{0}}^{\left(st\right)} & =\;\int \int \;dqdp\frac{e^{-\beta H_{0}}}{Z_{0}}p=\frac{\int_{-\infty }^{\infty }pe^{-\frac{\beta p^{2}}{2}}dp\int_{-\infty }^{\infty }e^{-\frac{\beta \omega^{2}q^{2}}{2}}dq}{\;\int \int \;e^{-\beta H_{0}}dqdp}\\ \; & =\frac{1}{\int_{-\infty }^{\infty }e^{-\frac{\beta p^{2}}{2}}dp}\frac{-e^{-\frac{\beta p^{2}}{2}}}{\beta }{|}_{-\infty }^{\infty }=\frac{1}{\sqrt{2\pi /\beta }}\frac{-e^{-\frac{\beta p^{2}}{2}}}{\beta }{|}_{-\infty }^{\infty }=0, \end{array}$$
(105)$$\begin{array}{cc} {\langle p^{2}\rangle }_{f_{0}}^{\left(st\right)} & =\;\int \int \;dqdp\frac{e^{-\beta H_{0}}}{Z_{0}}p^{2}=\frac{\int_{-\infty }^{\infty }p^{2}e^{-\frac{\beta p^{2}}{2}}dp}{\int_{-\infty }^{\infty }e^{-\frac{\beta p^{2}}{2}}dp}\\ \; & =\frac{-\frac{p}{\beta }e^{-\frac{\beta p^{2}}{2}}{|}_{-\infty }^{\infty }+\int_{-\infty }^{\infty }\frac{1}{\beta }e^{-\frac{\beta p^{2}}{2}}}{\sqrt{2\pi /\beta }}=\frac{\frac{1}{\beta }\sqrt{2\pi /\beta }}{\sqrt{2\pi /\beta }}=k_{b}T, \end{array}$$
(106)$$\begin{array}{cc} {\langle q^{2}\rangle }_{f_{0}}^{\left(st\right)} & =\;\int \int \;dqdp\frac{e^{-\beta H_{0}}}{Z_{0}}q^{2}=\frac{\int_{-\infty }^{\infty }q^{2}e^{-\frac{\beta \omega^{2}q^{2}}{2}}dq}{\int_{-\infty }^{\infty }e^{-\frac{\beta \omega^{2}q^{2}}{2}}dq}\\ \; & =\frac{-\frac{q}{\beta \omega^{2}}e^{-\frac{\beta \omega^{2}q^{2}}{2}}{|}_{-\infty }^{\infty }+\int_{-\infty }^{\infty }\frac{1}{\beta \omega^{2}}e^{-\frac{\beta \omega^{2}q^{2}}{2}}}{\sqrt{2\pi /\beta \omega^{2}}}=\frac{\frac{1}{\beta \omega^{2}}\sqrt{2\pi /\beta \omega^{2}}}{\sqrt{2\pi /\beta \omega^{2}}}=\frac{k_{b}T}{\omega^{2}}. \end{array}$$
Then, the linear response for a single realization of the perturbation, i.e., for a given value *Z*, yields the following:(107)$$\begin{array}{cc} {\langle q\rangle }_{t}^{\left(st\right)} & ={\langle q\rangle }_{0}^{\left(st\right)}+\beta \int_{0}^{t}F{\langle \left(q\circ S_{0}^{s}\right)\cdot \dot{A}\rangle }_{0}^{\left(st\right)}\;ds\\ \; & =\frac{FZ}{2\omega }\left(\frac{\sin\left[(\omega -\gamma )t\right]}{\left(\omega -\gamma \right)}-\frac{\sin\left[(\omega +\gamma )t\right]}{\left(\omega +\gamma \right)}\right) \end{array}$$
On the other hand, the exact response is expressed by the following:(108)$$\begin{array}{cc} {\langle q\rangle }_{{\tilde{f}}_{t}}^{\left(st\right)} & ={\langle q\rangle }_{{\tilde{f}}_{0}}^{\left(st\right)}+\int_{0}^{t}{\langle \left(q\circ S_{w}^{s}\right)\cdot {\tilde{\Omega }}^{f_{0}}\rangle }_{{\tilde{f}}_{0}}^{\left(st\right)}\;ds=\int_{0}^{t}{\langle \left(q\circ S_{w}^{s}\right)\cdot \frac{p}{k_{b}T}w\left(\theta ,\phi \right)\rangle }_{{\tilde{f}}_{0}}^{\left(st\right)}\;ds\\ \; & =\int_{0}^{t}ds\int d\theta d\phi dqdp\;w\left(\theta ,\phi \right)\frac{1}{\sigma }\frac{e^{-H_{0}/\left(k_{b}T\right)}}{Z_{0}}\frac{p}{k_{b}T}[\left(q+FZ\frac{\phi }{\gamma (\omega^{2}-\gamma^{2})}\right)\cos\left(\omega s\right)\\ \; & \;+\frac{1}{\omega }\left(p-FZ\frac{\gamma \theta }{\left(\omega^{2}-\gamma^{2}\right)}\right)\sin\left(\omega s\right)+\frac{FZ}{\left(\omega^{2}-\gamma^{2}\right)}\left(\theta \sin\left(\gamma s\right)-\frac{\phi }{\gamma }\cos\left(\gamma s\right)\right)]\\ \; & =\int_{0}^{t}ds\int d\theta d\phi dqdp\;\frac{w(\theta ,\phi )}{k_{b}T}\frac{1}{\sigma }\frac{e^{-H_{0}/\left(k_{b}T\right)}}{Z_{0}}\frac{p^{2}}{\omega }\sin\left(\omega s\right)\\ \; & =-\frac{FZ}{\sigma \gamma }\int_{0}^{t}ds\frac{\sin\left(\omega s\right)}{\omega }\;\int \int \;d\theta d\phi \left(-\gamma \theta \sin\left(\gamma s\right)+\phi \cos\left(\gamma s\right)\right)=0 \end{array}$$
because the integral in θ and ϕ vanishes (see Appendix A for details). Therefore, the linear and exact responses differ under the chosen conditions. If, on the other hand, the initial condition of the auxiliary variables is fixed to a single point with $\theta_{0}=1$ and $\phi_{0}=0$, we have F=1, and
(109)$$\tilde{f_{0}}\left(\Gamma ,\theta ,\phi \right)=\delta \left(\theta_{0}-1\right)\delta \left(\phi_{0}\right)f_{0}\left(\Gamma \right)=\delta \left(\theta_{0}-1\right)\delta \left(\phi_{0}\right)\frac{e^{-\beta H_{0}}}{Z_{0}}$$
The exact response is then expressed by the following:(110)$$\begin{array}{cc} {\langle q\rangle }_{{\tilde{f}}_{t}}^{\left(st\right)} & ={\langle q\rangle }_{{\tilde{f}}_{0}}^{\left(st\right)}+\int_{0}^{t}{\langle \left(q\circ S_{w}^{s}\right)\cdot \tilde{\Omega^{f_{0}}}\rangle }_{{\tilde{f}}_{0}}^{\left(st\right)}\;ds=\int_{0}^{t}{\langle \left(q\circ S_{w}^{s}\right)\cdot \frac{p}{k_{b}T}w\left(\theta ,\phi \right)\rangle }_{{\tilde{f}}_{0}}^{\left(st\right)}\;ds\\ \; & =\int_{0}^{t}ds\int d\theta d\phi dqdp\;w\left(\theta ,\phi \right)\delta \left(\theta -1\right)\delta \left(\phi \right)\frac{e^{-H_{0}/\left(k_{b}T\right)}}{Z_{0}}\frac{p}{k_{b}T}[\left(q+Z\frac{\phi }{\gamma (\omega^{2}-\gamma^{2})}\right)\cos\left(\omega s\right)\\ \; & \;+\frac{1}{\omega }\left(p-Z\frac{\gamma \theta }{\left(\omega^{2}-\gamma^{2}\right)}\right)\sin\left(\omega s\right)+\frac{Z}{\left(\omega^{2}-\gamma^{2}\right)}\left(\theta \sin\left(\gamma s\right)-\frac{\phi }{\gamma }\cos\left(\gamma s\right)\right)]\\ \; & =\int_{0}^{t}ds\int d\theta d\phi dqdp\;\frac{w(\theta ,\phi )}{k_{b}T}\delta \left(\theta -1\right)\delta \left(\phi \right)\frac{e^{-H_{0}/\left(k_{b}T\right)}}{Z_{0}}\frac{p^{2}}{\omega }\sin\left(\omega s\right)\\ \; & =-\frac{Z}{\gamma }\int_{0}^{t}ds\frac{\sin\left(\omega s\right)}{\omega }\;\int \int \;d\theta d\phi \;\delta \left(\theta -1\right)\delta \left(\phi \right)\left(-\gamma \theta \sin\left(\gamma s\right)+\phi \cos\left(\gamma s\right)\right)\\ \; & =Z\int_{0}^{t}ds\frac{\sin\left(\omega s\right)}{\omega }\sin\left(\gamma s\right)=\frac{Z}{2\omega }\left(\frac{\sin\left[(\omega -\gamma )t\right]}{\left(\omega -\gamma \right)}-\frac{\sin\left[(\omega +\gamma )t\right]}{\left(\omega +\gamma \right)}\right) \end{array}$$
where the integral on the curve in dθ and dϕ gives the length of the curve that is σ. This is the same as the linear response with $\left(\theta_{0},\phi_{0}\right)$ uniformly distributed in the unit circle. If we take O=p as an observable, we obtain something similar. For the linear response, we have the following:(111)$$\begin{array}{cc} {\langle p\rangle }_{t}^{\left(st\right)} & ={\langle p\rangle }_{0}^{\left(st\right)}+\beta \int_{0}^{t}F\left(s\right){\langle \left(p\circ S_{0}^{s}\right)\cdot \dot{A}\rangle }_{0}^{\left(st\right)}\;ds\\ \; & =FZ\int_{0}^{t}ds\sin\left(\gamma s\right)\;\int \int \;dqdp\frac{e^{-H_{0}/\left(k_{b}T\right)}}{Z_{0}}\frac{p}{k_{b}T}\left[-\omega q\sin\left(\omega s\right)+p\cos\left(\omega s\right)\right]\\ \; & =FZ\int_{0}^{t}ds\sin\left(\gamma s\right)\;\int \int \;dqdp\frac{e^{-H_{0}/\left(k_{b}T\right)}}{Z_{0}}\frac{p^{2}}{k_{b}T}\cos\left(\omega s\right)\\ \; & =FZ\int_{0}^{t}ds\sin\left(\gamma s\right)\cos\left(\omega s\right)=\frac{\epsilon Z}{2}\int_{0}^{t}\left[\sin\left[(\gamma -\omega )s\right]+\sin\left[(\gamma +\omega )s\right]\right]ds\\ \; & =\frac{FZ}{2}\left(-\frac{\cos\left[(\gamma -\omega )t\right]}{\left(\gamma -\omega \right)}-\frac{\cos\left[(\gamma +\omega )t\right]}{\left(\gamma +\omega \right)}+2\right) \end{array}$$
and for the exact response, we have the following:(112)$$\begin{array}{cc} {\langle p\rangle }_{{\tilde{f}}_{t}}^{\left(st\right)} & ={\langle p\rangle }_{{\tilde{f}}_{0}}^{\left(st\right)}+\int_{0}^{t}{\langle \left(p\circ S_{w}^{s}\right)\cdot {\tilde{\Omega }}^{f_{0}}\rangle }_{{\tilde{f}}_{0}}^{\left(st\right)}\;ds=\int_{0}^{t}{\langle \left(p\circ S_{w}^{s}\right)\cdot \frac{p}{k_{b}T}w\left(\theta ,\phi \right)\rangle }_{{\tilde{f}}_{0}}^{\left(st\right)}\;ds\\ \; & =\int_{0}^{t}ds\int d\theta d\phi dqdp\;w\left(\theta ,\phi \right)\frac{1}{\sigma }\frac{e^{-H_{0}/\left(k_{b}T\right)}}{Z_{0}}\frac{p}{k_{b}T}[-\omega \left(q+FZ\frac{\phi }{\gamma (\omega^{2}-\gamma^{2})}\right)\sin\left(\omega s\right)\\ \; & \;+\left(p-FZ\frac{\gamma \theta }{\left(\omega^{2}-\gamma^{2}\right)}\right)\cos\left(\omega s\right)+\frac{FZ}{\left(\omega^{2}-\gamma^{2}\right)}\left(\theta \gamma \cos\left(\gamma s\right)+\phi \sin\left(\gamma s\right)\right)]\\ \; & =\int_{0}^{t}ds\int d\theta d\phi dqdp\;\frac{w(\theta ,\phi )}{k_{b}T}\frac{1}{\sigma }\frac{e^{-H_{0}/\left(k_{b}T\right)}}{Z_{0}}\frac{p^{2}}{\omega }\cos\left(\omega s\right)\\ \; & =-\frac{FZ}{\sigma \gamma }\int_{0}^{t}ds\frac{\cos\left(\omega s\right)}{\omega }\;\int \int \;d\theta d\phi \left(-\gamma \theta \sin\left(\gamma s\right)+\phi \cos\left(\gamma s\right)\right)=0 \end{array}$$
Again, the same response is obtained only for the fixed initial condition $\left(\theta_{0}=1,\phi_{0}=0\right)$. This is not strange; it happens when the exact response amounts to a first order perturbation, which is the case in special situations [39]. However, in general, this condition is not verified. For instance, we consider the observable O=qp, and the linear theory yields the following:(113)$$\begin{array}{cc} {\langle qp\rangle }_{f_{t}}^{\left(st\right)} & ={\langle qp\rangle }_{f_{0}}^{\left(st\right)}+\beta \int_{0}^{t}F\left(s\right){\langle \left(q\circ S_{0}^{s}\right)\left(p\circ S_{0}^{s}\right)\cdot \dot{A}\rangle }_{f_{0}}^{\left(st\right)}\;ds\\ \; & =FZ\int_{0}^{t}ds\sin\left(\gamma s\right)\;\int \int \;dqdp\frac{e^{-H_{0}/\left(k_{b}T\right)}}{Z_{0}}\frac{p}{k_{b}T}[\left(q\cos\left(\omega s\right)+\frac{p}{\omega }\sin\left(\omega s\right)\right)\\ \; & \;\times \left(-\omega q\sin\left(\omega s\right)+p\cos\left(\omega s\right)\right)]=0 \end{array}$$
because integrating the terms $pq^{2}$, $qp^{2}$, $p^{3}$, over dqdp yields zero. For the exact response, we have the following:(114)$$\begin{array}{cc} {\langle qp\rangle }_{{\tilde{f}}_{t}} & ={\langle qp\rangle }_{{\tilde{f}}_{0}}+\int_{0}^{t}ds{\langle w\left(\theta ,\phi \right)\frac{p}{k_{b}T}\left(S_{w}^{S}\circ q\right)\left(S_{w}^{s}\circ p\right)\rangle }_{{\tilde{f}}_{0}}\\ \; & =\int_{0}^{t}ds\langle w\left(\theta ,\phi \right)\frac{p}{k_{b}T}\{[C_{1}\cos\left(\omega t\right)+\frac{C_{2}}{\omega }\sin\left(\omega t\right)+\frac{\epsilon Z}{\left(\omega^{2}-\gamma^{2}\right)}(\theta_{0}\sin\left(\gamma t\right)+\\ \; & {-\frac{\phi_{0}}{\gamma }\cos\left(\gamma t\right))]\left[-\omega C_{1}\sin\left(\omega t\right)+C_{2}\cos\left(\omega t\right)+\frac{\epsilon Z}{\left(\omega^{2}-\gamma^{2}\right)}\left(\theta_{0}\gamma \cos\left(\gamma t\right)+\phi_{0}\sin\left(\gamma t\right)\right)\right]\}\rangle }_{{\tilde{f}}_{0}} \end{array}$$
Taking into account all the terms that cancel under integration over dqdp, and those that cancel by integrating over dθdϕ, what remains are elliptic integrals over dθdϕ, which can be solved numerically, and integrals over *t* of sine and cosine terms. The result does not vanish, as can be seen by inspection. In this example, even fixing the initial conditions as $\left(\theta_{0}=1,\phi_{0}=0\right)$ and using the Dirac delta distribution, we obtain a non-vanishing result:(115)$$\begin{array}{ccc} & & {\langle qp\rangle }_{{\tilde{f}}_{t}}^{\left(st\right)}=\\ & & -\frac{FZ}{\gamma }\int_{0}^{t}ds\frac{-2FZ\gamma }{\omega (\omega^{2}-\gamma^{2})}\sin\left(\omega s\right)\cos\left(\omega s\right)\;\int \int \;d\theta d\phi \delta \left(\theta -1\right)\delta \left(\phi \right)\left(-\gamma \theta \sin\left(\gamma s\right)+\phi \cos\left(\gamma s\right)\right)\\ & & -\frac{FZ}{\gamma }\int_{0}^{t}ds\frac{FZ\gamma }{\omega (\omega^{2}-\gamma^{2})}\cos\left(\gamma s\right)\;\int \int \;d\theta d\phi \;\delta \left(\theta -1\right)\delta \left(\phi \right)\left(-\gamma \theta \sin\left(\gamma s\right)+\phi \cos\left(\gamma s\right)\right)\\ & & =-\frac{2F^{2}Z^{2}}{\omega (\omega^{2}-\gamma^{2})}\left[\frac{\left(\gamma -2\;\omega \right)\;\sin\left(\gamma \;t+2\;\omega \;t\right)}{4\;\gamma^{2}-16\;\omega^{2}}-\frac{\left(\gamma +2\;\omega \right)\;\sin\left(\gamma \;t-2\;\omega \;t\right)}{4\;\gamma^{2}-16\;\omega^{2}}+\frac{\sin^{2}\gamma t}{4\gamma }\right] \end{array}$$
In this case, the linear response yields zero because the exact response is second order in the perturbation magnitude F. When that is sufficiently small, the terms of order $O(F^{2})$ can be neglected, but not in general. Furthermore, forced oscillators may undergo resonance phenomena, greatly amplifying the amplitude of their oscillations even if they are driven by very weak forces. This is captured by the denominators of expression (Equation 116), which are missing in the linear response. Near the resonances, the linear response is bound to fail, even with small driving forces, and the exact formula is necessary.

## 6. Viscous Medium and General Stochastic Forcing

As the unperturbed dynamics, consider the one-dimensional motion of a single particle in a viscous fluid of constant viscosity γ:(116)$$\dot{x}=v\;;\;\dot{v}=-\gamma v$$
It is irrelevant that this is not a Hamiltonian system, as the theory illustrated above applies to general dynamical systems. As the distribution of the initial conditions, let us take, for simplicity,
(117)$$f_{0}\left(x,v\right)=\frac{1}{2\pi }\exp\left(-\frac{x^{2}+v^{2}}{2}\right)$$
which is not invariant for the unperturbed dynamics and implies the following:(118)$$\Omega_{f_{0}}\left(x,v\right)=\gamma +xv-\gamma v^{2}$$
As $f_{0}$ is not invariant under the unperturbed dynamics that, in turn, are not Hamiltonian, the dissipation function does not need to represent the dissipated power. However, the mathematics can still be performed.

Let us perturb the dynamics with stochastic forcing of mean F>0:(119)$$\dot{x}=v\;;\;\dot{v}=-\gamma v+F+\sum_{n=-\infty }^{\infty }\alpha_{n}w_{n}\left(\theta ,\phi \right)\;;\;\dot{\theta }=\phi \;;\;\dot{\phi }=-\theta$$
so that the mean of each expansion coefficient vanishes. If the initial distribution of the phases (θ,ϕ) is uniform in the unit circle, $g_{0}\left(\theta ,\phi \right)=1/2\pi$, the initial distribution on the enlarged phase space takes the following form:(120)$${\tilde{f}}_{0}\left(x,v,\theta ,\phi \right)=\frac{1}{4\pi^{2}}\exp\left(-\frac{x^{2}+v^{2}}{2}\right)$$
which implies
(121)$$\frac{\partial g_{0}}{\partial \theta }=0=\frac{\partial g_{0}}{\partial \phi }$$
In this example, the original phase space has two dimensions, Γ=(x,v), and the perturbation only appears in the equation for *v*. Therefore, we have $\alpha_{n}=\left(\alpha_{nx},\alpha_{nv}\right)=\left(0,\alpha_{nv}\right)$. Moreover, $w_{n}$ only depends on θ, because in the unit circle, we have ϕ=ϕ(θ). We can then write
(122)$${\tilde{\Omega }}_{{\tilde{f}}_{0}}\left(x,v,\theta ,\phi \right)=\gamma +xv-\gamma v^{2}+Fv+\sum_{n=-\infty }^{\infty }v\;\alpha_{nv}\;W_{n}\left(\theta \right)$$
where $W_{n}\left(\theta \right)$ represents $w_{n}\left(\theta ,\phi \right)$ in the unit circle. Consequently, the average at time *t* of an observable O takes the following form:(123)$$\begin{array}{ccc} {\langle O\rangle }_{{\tilde{f}}_{t}}\; & = & \;{\langle O\rangle }_{f_{0}}+\frac{1}{4\pi^{2}}\int_{0}^{t}ds\int_{R^{2}}dx\;dv\int_{0}^{2\pi }d\theta \times \\ \; & \times & \;\;\exp\left(-\frac{x^{2}+v^{2}}{2}\right)O\left(S^{s}\left(x,v\right)\right)\left[\gamma +xv-\gamma v^{2}+Fv+\sum_{n=-\infty }^{\infty }v\;\alpha_{nv}\;W_{n}\left(\theta \right)\right]\\ \; & = & \;{\langle O\rangle }_{f_{t}}+\int_{0}^{t}ds\int_{R^{2}}\frac{dx\;dv}{4\pi^{2}}\;v\;e^{-\frac{x^{2}+v^{2}}{2}}\int_{0}^{2\pi }d\theta \;O\left(S^{s}\left(x,v\right)\right)\left[F+\sum_{n=-\infty }^{\infty }\alpha_{nv}\;W_{n}\left(\theta \right)\right] \end{array}$$
which can be computed once the distribution of the vectors $\alpha_{n}$ is given, considering that the evolution $S^{s}\left(x,v\right)$ also depends on θ, while $\alpha_{n}$ and (x,v) do not. The performance of this formula, compared to direct averaging, must be assessed numerically, and is expected to be superior to direct averaging, as in the time-independent deterministic case, because the convergence issues one expects are analogous.

## 7. Concluding Remarks

In this paper, we provided a generalization to stochastic processes of the exact response theory developed for deterministic molecular dynamics models, also known as TTCF in the field of molecular dynamics, where it mostly concerns time-independent perturbations, apart from a few seminal papers on periodic perturbations. This was conducted, starting from the previously developed time-dependent response theory of Ref. [15], expressing the stochastic perturbation as a Karhunen–Loève expansion, and then adding two auxiliary variables that make the corresponding time-dependent system of ordinary differential equations autonomous. As in Ref. [15], which is based on the Fourier expansion of the perturbing field, information on spectral properties of the forcing and response are directly available. The main ingredient of the exact response theory, the dissipation function, has then been derived in the enlarged phase space. This allows us to treat processes whose initial conditions are variously distributed, adding flexibility to the general response theory, which considers fixed initial perturbations. For instance, we can now compute different kinds of averages, which describe different experimental situations, i.e., time averages over single realizations of stochastic processes, ensemble averages over the initial conditions in the enlarged phase space, and combinations of the two.

We illustrated the theory by applying it to a one-dimensional harmonic oscillator, perturbed by a sinusoidal force, whose amplitude can be either deterministic or stochastic. This is enough to compare the exact response with the linear response. Although in special situations, the two responses may coincide to a certain degree, defining the range of applicability of the linear theory, they differ in general, even in such simple situations. The exact formula results are necessary, not only in the presence of large perturbations, as obvious, but also for small perturbations, in the presence of resonance phenomena. Furthermore, we have shown that different results are obtained for different distributions of the initial perturbation, something that is not obvious within the standard linear response theory. Indeed, enlarging the phase space to make the system autonomous allows us to choose the distribution of the initial condition of the perturbation, referring to different kinds of experiments, where the initial perturbation can either be deterministically fixed, as commonly assumed, or distributed in various ways.

As a second simple example, we considered one particle moving in a one-dimensional viscous medium, under the action of stochastic driving with a non-vanishing mean, so that a non-equilibrium stationary state may eventually be reached. Note that, even in this case of perturbations, which exert a net action on the system, we needed to explicitly develop the theory for zero mean perturbations only. Indeed, the possible net perturbation may be included in the autonomous part of the dynamics, for which the exact theory was developed in the past; the results are applicable to generic dynamical systems, extending beyond the field of molecular dynamics where it was born.

The above is in addition to the fact that many theoretical results can be derived from the exact response formula before they can be numerically computed in simulations, e.g., see [7,40,41]. Moreover, the linear response cannot treat phase transitions [42], and various studies show an impressively better performance of the TTCF formula compared to other averaging methods for time-independent perturbations [5,6]. We expect this to be the case with our approach when dealing with time-dependent (deterministic or stochastic) perturbations. This will constitute the basis of future works.

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
