# Peer review of "Exact Response Theory for Time-Dependent and Stochastic Perturbations"

_entropy, 2023, doi:10.3390/e26010012_

Round 1
Reviewer 1 Report
Comments and Suggestions for Authors
In this paper the authors extend a previous work on time dependent nonlinear response theory and apply it to stochastic systems so that it is now generalised to both deterministic and stochastic systems. They then apply the theory to the case of a harmonic oscillator and deduce both the linear and full (exact) nonlinear response for arbitrary phase variables and demonstrate the importance of the exact nonlinear response, which is able to account for all subtleties which a naïve implementation of linear response theory would be incapable of detecting.
I find this an enlightening and comprehensive piece of work that adds further tools to the growing armoury of exact nonlinear response theory. Its main contribution is to demonstrate that the generalised framework of the transient-time correlation function (TTCF) methodology can be extended to include stochastic systems, not just deterministic ones. This is a significant advance, and the authors should be commended for this. I have a few suggestions and comments for the authors before publication can be recommended.
1. The generalisation to time dependent fields was done in a previous paper, reference 11, of which the second author is also an author on (incidentally, the journal name is missing from this reference; please include “Symmetry” as the journal name). In both that paper and this current manuscript, the theory seems applicable to a time dependence that is implicitly periodic in nature. It is this periodicity that allows the authors to remove the explicit time dependence and thus convert the system of equations to ones that are autonomous. The authors may not be aware that a similar formulation for periodic time dependence was done by Petravic and Evans for nonequilibrium (deterministic) MD simulations of particles under time-periodic planar shear flow (J. Petravic and D. J. Evans, Phys. Rev. Lett. 78, 1199 (1997); J. Petravic and D. J. Evans, Phys. Rev. E 56, 1207 (1997)), and then by Todd for time-periodic planar elongational flow (B. D. Todd, Phys. Rev. E 58, 4587 (1998)). Although this was done before the generalisation of TTCF to incorporate the dissipation function formalism, it is likely that these prior works are similar in nature to what the authors have done. The authors have certainly generalised the procedure further and have demonstrated its applicability to stochastic systems. It would be good if the authors can comment on how these previous works relate to their formalism.
2. Following from the above observation, can the authors comment on the possibility (or not) of developing a time-independent response theory that is truly general and does not require a system to be time-periodic?
3. On page 6 the authors write “Eq (35) concerns the correlation function of the evolving observable of interest with the dissipative flux and such correlation function is an average computed with respect to the initial distribution f0.” I am a little confused by this. It gives the conception that the correlation function is done in the equilibrium ensemble, but clearly it is not, as the authors say in the following two sentences. I do not understand why the integral in Eq (35) seems to be done in the equilibrium ensemble? Eq (35), as shown in reference 10, is the exact nonlinear response which can obtained by time integrating the differential form of the time evolving phase variable, as is done in the book on NEMD by Todd and Daivis (note a typographical error in their Eq (3.60) – 2 lines above the final result, f0 should be a function of t, not evaluated at t = 0). As such, shouldn’t the ensemble average in the RHS of Eq (35) be the ensemble average in the nonequilibrium system, i.e. the integration over the full phase space of the nonequilibrium distribution is assumed? Perhaps I am misunderstanding something here, so could the authors please clarify this a bit more in this section?
4. While I appreciate the analytic solution to the linear and nonlinear response for the harmonic oscillator, I feel the best demonstration of the power and value of the theory would be to apply it to a stochastic simulation, such as a system of particles undergoing nonequilibrium Langevin or Brownian dynamics. The authors could then prove that the TTCF calculation of any phase variable is identical within statistical error to the direct time average. Also, they could show that the direct time average statistics would be inferior to that of the TTCF calculation for field strengths that approach those of physically meaningful measurements (i.e. laboratory measurements). This is the motivation for using TTCF, namely at realistic field strengths the signal to noise ratio will always be superior to that of direct time averaged results. This is true for nonequilibrium MD, and it should remain true for a system driven out of equilibrium in a nonequilibrium Langevin or Brownian dynamics simulation. To my knowledge, this has not yet been demonstrated in actual simulations.
Comments on the Quality of English Language
There are just a few minor grammatical errors. They are inconsequential because the paper is very well written overall. It is up to the authors if they want to carefully revise the grammar.
Author Response
Reply to Referee report 1
We would like to thank the Referee, for appreciating the purpose and significance of our work, and for insightful and constructive remarks, that allowed us to sensibly improve our manuscript. Below, we give the details.
The Referee states:
“In this paper the authors extend a previous work on time dependent nonlinear response theory and apply it to stochastic systems so that it is now generalised to both deterministic and stochastic systems. They then apply the theory to the case of a harmonic oscillator and deduce both the linear and full (exact) nonlinear response for arbitrary phase variables and demonstrate the importance of the exact nonlinear response, which is able to account for all subtleties which a naive implementation of linear response theory would be incapable of detecting.
I find this an enlightening and comprehensive piece of work that adds further tools to the growing armoury of exact nonlinear response theory. Its main contribution is to demonstrate that the generalised framework of the transient-time correlation function (TTCF) methodology can be extended to include stochastic systems, not just deterministic ones. This is a significant advance, and the authors should be commended for this. I have a few suggestions and comments for the authors before publication can be recommended.”
We are grateful to the Referee for such encouraging remarks, that highlight the purpose and significance of our work, acknowledging the novelty brought by our approach to stochastic perturbations.
The Referee states:
“1. The generalisation to time dependent fields was done in a previous paper, reference 11, of
which the second author is also an author on (incidentally, the journal name is missing from this reference; please include‚ ‘Symmetry’, as the journal name).”
Thank you. The reference has been fixed.
The Referee states:
“In both that paper and this current manuscript, the theory seems applicable to a time dependence that is implicitly periodic in nature. It is this periodicity that allows the authors to remove the explicit time dependence and thus convert the system of equations to ones that are autonomous. The authors may not be aware that a similar formulation for periodic time dependence was done by Petravic and Evans for nonequilibrium (deterministic) MD simulations of particles under time-periodic planar shear flow (J. Petravic and D. J. Evans, Phys. Rev. Lett. 78, 1199 (1997); J. Petravic and D. J. Evans, Phys. Rev. E 56, 1207 (1997)), and then by Todd for time-periodic planar elongational flow (B. D. Todd, Phys. Rev. E 58, 4587 (1998)). Although this was done before the generalisation of TTCF to incorporate the dissipation function formalism, it is likely that these prior works are similar in nature to what the authors have done.”
We are grateful to the Referee for this remark and for pointing out the references to earlier works, which are indeed important and precursors of our own work. We have added these papers to the Bibliography of our manuscript.
The Referee states:
“The authors have certainly generalised the procedure further and have demonstrated its applicability to stochastic systems. It would be good if the authors can comment on how these previous works relate to their formalism.”
The Referee is right. In some sense, everything was present in the early works on the TTCF, including those concerning periodic forcing. On the other hand, we have adopted a different perspective, both in the deterministic and in the stochastic cases, which allowed us to study aspects of response theory, not part of the earlier works, as far as we understand them. For instance, our work makes explicit that the original theory can be applied, with obvious adjustments,to generic dynamical systems, and not just to one class of molecular dynamics systems. This allows the use of the TTCF well beyond physics and the hard sciences. Also, our approach does not alter the time reversibility of the unperturbed dynamics, which is not obvious in non-autonomous systems. While this could be fixed in some non-autonomous dynamics, our approach makes reversibility directly applicable, and various results of statistical mechanics, relying on such a dynamical property, immediately follow. Finally, our approach makes evident the simplicity or complexity of the forcing, as well as of the response, through the identification of the contributions of the various harmonics to the corresponding signals. The resulting power spectra add useful information to the treatment of response, as common in statistical physics.
In the present version of our paper, this has been more clearly spelled out.
The Referee states:
“2. Following from the above observation, can the authors comment on the possibility (or not) of developing a time-independent response theory that is truly general and does not require a system to be time-periodic?”
Indeed, this is another important remark, which allowed us to clarify a fundamental aspect of our work. The answer is yes, and can be realized from two equivalent points of view. In the first place, as usually done in the Physics literature, the period of the perturbation can be taken as large as one wants, including much larger than any relevant time scale. This allows the resulting time dependence to be as complex as one may wish, in the time windows of physically realizable observations. Furthermore, the Fourier series mathematical formalism naturally extends to non-periodic signals, by letting the period tend to infinity. In that case, the Fourier series will turn, under relatively mild conditions, into a Fourier transform, suitable to treat non-periodic signals.
We have clarified this fact in the present version of our manuscript.
The Referee states:
“3. On page 6 the authors write ‘Eq (35) concerns the correlation function of the evolving observable of interest with the dissipative flux and such correlation function is an average computed with respect to the initial distribution f0’ I am a little confused by this. It gives the conception that the correlation function is done in the equilibrium ensemble, but clearly it is not, as the authors say in the following two sentences. I do not understand why the integral in Eq (35) seems to be done in the equilibrium ensemble? Eq (35), as shown in reference 10, is the exact nonlinear response which can obtained by time integrating the differential form of the time evolving phase variable, as is done in the book on NEMD by Todd and Daivis (note a typographical error in their Eq (3.60)‚ 2 lines above the final result, f0 should be a function of t, not evaluated at t = 0). As such, shouldn’t the ensemble average in the RHS of Eq (35) be the ensemble average in the nonequilibrium system, i.e. the integration over the full phase space of the nonequilibrium distribution is assumed? Perhaps I am misunderstanding something here, so could the authors please clarify this a bit more in this section?”
This is a very important remark, which allowed us to clarify one of the main useful features of the dissipation function formalism. The fact is that, indeed, the correlation function is to be computed with respect to the known (unperturbed) initial distribution, which makes it very easy to handle, compared to other response formulae. This is in the derivations that we do not report, because they ca ben found in numerous other works in the literature. On the other hand, the readers may be puzzled by such a powerful but counterintuitive result, and the notation used may at times leave doubts about what is really going one. For instance, Eq.(17) of Ref. [10] (now Ref.[13]), shows angular brackets with both driving field and probability density as subscripts, inside the time integral. Following the calculations, one realizes that the field is used for the dynamics, while the probability is the initial equilibrium ensemble, and is used to sample the initial conditions that are run with the nonequilibrium dynamics. The same can be evinced from Eq.(3.60) of the book by Todd and Daivis. The first step contains the time evolved probability density, but after differentiation the angular brackets are defined as averaging with respect to the initial distribution, cf. the symbol “≡”. In our notation, there is no ambiguity, as noted by the Referee.
Because this is both an important and delicate point, we have extended our discussion, of the exact response formula.
The Referee states:
“4. While I appreciate the analytic solution to the linear and nonlinear response for the harmonic oscillator, I feel the best demonstration of the power and value of the theory would be to apply it to a stochastic simulation, such as a system of particles undergoing nonequilibrium Langevin or Brownian dynamics. The authors could then prove that the TTCF calculation of any phase variable is identical within statistical error to the direct time average. Also, they could show that the direct time average statistics would be inferior to that of the TTCF calculation for field strengths that approach those of physically meaningful measurements (i.e. laboratory measurements). This is the motivation for using TTCF, namely at realistic field strengths the signal to noise ratio will always be superior to that of direct time averaged results. This is true for nonequilibrium MD, and it should remain true for a system driven out of equilibrium in a nonequilibrium Langevin or Brownian dynamics simulation. To my knowledge, this has not yet been demonstrated in actual simulations.”
We totally agree with the referee. The TTCF should prove superior to direct averaging also in the case of stochastic perturbations. This will be evidenced by numerical simulations of interesting systems, particularly nonequilibrium molecular dynamics. It is not the purpose of our present paper to present such numerical simulations, but thanks to the Referee’s remark, we have analyzed one simple nonequilibrium model: one particle under stochastic driving, with non zero mean. This was needed and improves the presentation of our theory.
List of modifications (all reported in blue font)
- to stress the importance of the TTCF, this sentence has been added in page 2: “and providing a superior method with respect to direct averaging, for such calculations”
- the central part of page 2 has been largely rewritten, to take into account Referee 1 remark on the deterministic time dependent theory previously developed by one of the authors and his collaborator.
- beginning of page 3 has been rewritten to describe the explicitly formulated stochastic example that has been introduced, following the Referee 1’s suggestion.
- the sentence “The Appendix computes some integrals used in the main text” has been added at the end of Section 1.
- (28) and the following sentence have been rewritten to improve readability, as required by Referee 2.
- the paragraph below Eq.(35) has been modified to stress the fact that the correlation function is computed with respect to the initial ensemble, and that the dynamics are the perturbed ones.
- the paragraph containing Eqs. (51) and (52) has been rewritten, to clarify the meaning of the various symbols used there, giving explicit definitions, as required by Referee 2.
- the title of Section 5 and the sentence below Eq. (91) have been modified, for further clarity.
- Section 6 has been added, to treat a simple example of stochastic perturbation.
- the Acnowledgements have been modified.
- the concluding have been partly rewritten to take into account the new presentation of results.
- the literature has been expanded, and one reference amended as suggested by Referee 2.
Reviewer 2 Report
Comments and Suggestions for Authors
The authors extend the exact response theory derived for deterministic systems to stochastic ones. Their approach of converting to autonomous dynamics in an extended phase space is particularly clever. They explore the consequences of different possible averaging methods, and find a different response in each case.
The manuscript is very clearly written, and the reasoning well explained. Inclusion of the harmonic oscillator example is very helpful in understanding the results and demonstrates the importance of going beyond linear response. I expect this work will drive innovation in molecular simulation methods.
A few minor suggested changes:
1. After equation (28), the text “and turns f_t after a time t” is not clear.
2. Line 137 is hard to follow because the definition of S tilde is hard to comprehend from this text.
3. Typo in line 220: [6?]
Author Response
Reply to Referee report 2
We are grateful to the Referee for encouraging remarks, and for pointing out aspects of our manuscript that needed improvement. We have carefully followed the Referee’s suggestions. Below we give the details.
The Referee states:
“The authors extend the exact response theory derived for deterministic systems to stochastic ones. Their approach of converting to autonomous dynamics in an extended phase space is particularly clever. They explore the consequences of different possible averaging methods, and find a different response in each case.
The manuscript is very clearly written, and the reasoning well explained. Inclusion of the harmonic oscillator example is very helpful in understanding the results and demonstrates the importance of going beyond linear response. I expect this work will drive innovation in molecular simulation methods.”
Many thanks for such encouraging words.
The Referee states:
A few minor suggested changes:
- After equation (28), the text “and turns f_t after a time t” is not clear.
We agree. To clarify, we have amended the sentence as follows: “We denote by f_t the probability density at time t, obtained as solution of Eq.(28) with initial condition f_0.”
The Referee states:
“2. Line 137 is hard to follow because the definition of S tilde is hard to comprehend
from this text.”
Indeed, we realize that our statement was not properly formulated. We have thuse rewritten the whole paragraph, explicitly defining the various terms that appear in Eqs. (51) and (52). We believe that this substantially improves the presentation of our results.
The Referee states:
“3. Typo in line 220: [6?]”
There was a typo indeed. It has been fixed.
List of modifications (all reported in blue font)
- to stress the importance of the TTCF, this sentence has been added in page 2: “and providing a superior method with respect to direct averaging, for such calculations”
- the central part of page 2 has been largely rewritten, to take into account Referee 1 remark on the deterministic time dependent theory previously developed by one of the authors and his collaborator.
- beginning of page 3 has been rewritten to describe the explicitly formulated stochastic example that has been introduced, following the Referee 1’s suggestion.
- the sentence “The Appendix computes some integrals used in the main text” has been added at the end of Section 1.
- (28) and the following sentence have been rewritten to improve readability, as required by Referee 2.
- the paragraph below Eq.(35) has been modified to stress the fact that the correlation function is computed with respect to the initial ensemble, and that the dynamics are the perturbed ones.
- the paragraph containing Eqs. (51) and (52) has been rewritten, to clarify the meaning of the various symbols used there, giving explicit definitions, as required by Referee 2.
- the title of Section 5 and the sentence below Eq. (91) have been modified, for further clarity.
- Section 6 has been added, to treat a simple example of stochastic perturbation.
- the Acnowledgements have been modified.
- the concluding have been partly rewritten to take into account the new presentation of results.
- the literature has been expanded, and one reference amended as suggested by Referee 2.
Round 2
Reviewer 1 Report
Comments and Suggestions for Authors
The authors have comprehensively addressed all concerns of the previous report.